# Empowering Collaborative Filtering with Principled Adversarial Contrastive Loss

**An Zhang**[1]* **Leheng Sheng**[2]* **Zhibo Cai**[3]† **Xiang Wang**[4]  **Tat-Seng Chua**[1]
[1]National University of Singapore
[2]Tsinghua University
[3]Center for Applied Statistics and School of Statistics, Renmin University of China
[4]University of Science and Technology of China
anzhang@u.nus.edu, chenglh22@mails.tsinghua.edu.cn,
caizhibo@ruc.edu.cn, xiangwang1223@gmail.com, dcscts@nus.edu.sg

## Abstract

Contrastive Learning (CL) has achieved impressive performance in self-supervised learning tasks, showing superior generalization ability. Inspired by the success, adopting CL into collaborative filtering (CF) is prevailing in semi-supervised top-$K$ recommendations. The basic idea is to routinely conduct heuristic-based data augmentation and apply contrastive losses (*e.g.,* InfoNCE) on the augmented views. Yet, some CF-tailored challenges make this adoption suboptimal, such as the issue of out-of-distribution, the risk of false negatives, and the nature of top-$K$ evaluation. They necessitate the CL-based CF scheme to focus more on mining hard negatives and distinguishing false negatives from the vast unlabeled user-item interactions, for informative contrast signals. Worse still, there is limited understanding of contrastive loss in CF methods, especially *w.r.t.* its generalization ability. To bridge the gap, we delve into the reasons underpinning the success of contrastive loss in CF, and propose a principled **Adv**ersarial **InfoNCE** loss (**AdvInfoNCE**), which is a variant of InfoNCE, specially tailored for CF methods. AdvInfoNCE adaptively explores and assigns hardness to each negative instance in an adversarial fashion and further utilizes a fine-grained hardness-aware ranking criterion to empower the recommender's generalization ability. Training CF models with AdvInfoNCE, we validate the effectiveness of AdvInfoNCE on both synthetic and real-world benchmark datasets, thus showing its generalization ability to mitigate out-of-distribution problems. Given the theoretical guarantees and empirical superiority of AdvInfoNCE over most contrastive loss functions, we advocate its adoption as a standard loss in recommender systems, particularly for the out-of-distribution tasks. Codes are available at https://github.com/LehengTHU/AdvInfoNCE.

## 1 Introduction

Contrastive Learning (CL) has emerged as a potent tool in self-supervised learning tasks [1–4], given its superior generalization ability. By simultaneously pulling positive pairs close together while pushing apart negative pairs in the feature space [5], CL has demonstrated the ability to extract general features from limited signals. This promising result has propelled research interest in leveraging CL for Collaborative Filtering (CF) in top-$K$ recommendation tasks, leading to a marked enhancement in performance and generalization ability [6–12]. Specifically, the prevalent paradigm in CL-based CF methods is to routinely adopt heuristic-based data augmentation for user-item bipartite graphs

---

*An Zhang and Leheng Sheng contribute equally to this work.
†Zhibo Cai is the corresponding author.

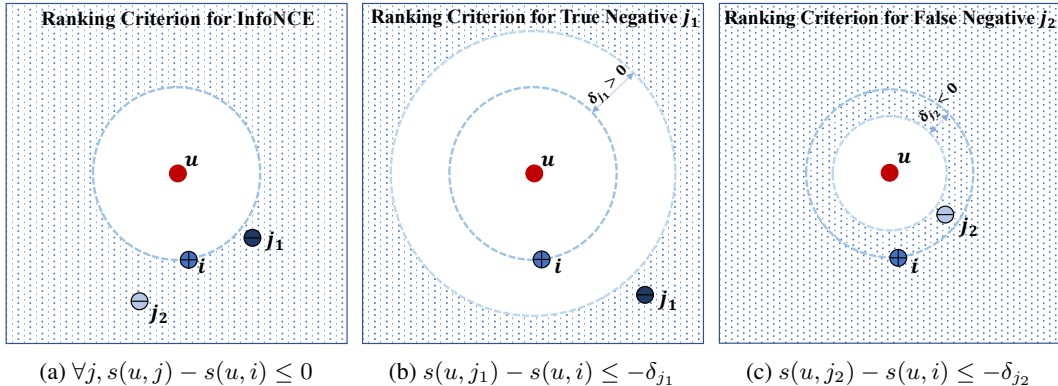

(a) $\forall j, s(u, j) - s(u, i) \leq 0$  (b) $s(u, j_1) - s(u, i) \leq -\delta_{j_1}$  (c) $s(u, j_2) - s(u, i) \leq -\delta_{j_2}$

Figure 1: Comparative geometric interpretation of ranking criteria based on similarity measure $s(\cdot, \cdot)$ for InfoNCE (1a) and our proposed AdvInfoNCE (1b-1c). The figures depict the user $u$ as a red dot, with items $i$ (positive), $j_1$ (true negative), and $j_2$ (false negative) depicted as points with varying shades of blue. The color gradient reflects the underlying similarity between user $u$ and the items, while the shaded region delineates the feasible zone for its corresponding negative item.

[13–15], coupled with the indiscriminate application of contrastive losses such as InfoNCE [16]. Wherein, a common assumption underlying these methods is considering all unobserved interactions as negative signals [17, 18].

Despite the empirical success of CL-based CF methods, two critical limitations have been observed that potentially impede further advancements in this research direction:

- **Lack of considering the tailored inductive bias for CF.** As a standard approach for semi-supervised top-$K$ recommendation with implicit feedback, CF methods face unique challenges. In most cases, the majority of user-item interactions remain unobserved and unlabelled, from which CF methods directly draw negative instances. However, indiscriminately treating these unobserved interactions as negative signals overlooks the risk of false negatives, thus failing to provide reliable contrastive signals [19]. This issue is known as the exposure bias [20, 21] in recommender systems. Unfortunately, this inherent inductive bias is rarely considered in current CL-based CF methods, resulting in suboptimal recommendation quality and generalization ability [22].

- **Limited theoretical understanding for the generalization ability of CL-based CF methods.** While data augmentation, as a key factor for generalization ability, is pivotal in computer vision [23, 24], CL-based CF models exhibit insensitivity to perturbations in the user-item bipartite graph [25]. Recent studies [26–28] empirically reveal that contrastive loss plays a more crucial role in boosting performance in CF than heuristic-based graph augmentation. However, an in-depth theoretical understanding of contrastive loss in top-$K$ recommendation that could shed light on the generalization ability of CL-based CF methods remains largely under-explored.

To better understand and reshape the contrastive loss specially tailored for top-$K$ recommendation, we aim to automatically and adversarially assign hardness to each negative user-item pair, which de facto concurrently enlarges the similarity discrepancy for hard negatives and loosens the ranking constraint for false negatives. To be specific, the fundamental idea revolves around discerning a fine-grained degree of hardness for negative interactions, thereby yielding more profound insight into the underlying ranking criterion for top-$K$ recommendation, as depicted in Figure 1. Notably, for a given user $u$, $\delta_j$ refers to the hardness of item $j$. When $\delta_{j_1} > 0$ (as seen in Figure 1b), item $j_1$ is identified as a hard negative, resulting in a significant contraction of the feasible representation space. Conversely, when $\delta_{j_2} < 0$ (as seen in Figure 1c), the ranking criterion is relaxed compared to the constraint of InfoNCE, allowing the recommender to mitigate the impact of false negative noise. Nevertheless, the tasks of effectively distinguishing between the hard and false negatives, and of learning a proper fine-grained ranking of interactions continue to present significant challenges [22].

To this end, we incorporate a fine-grained hardness-aware ranking criterion and devise a slightly altered version of InfoNCE loss through adversarial training. This modified contrastive loss coined **AdvInfoNCE**, is explicitly designed for robust top-$K$ recommendation. Specifically, we frame hardness learning as an adversarial optimization problem by designing a hardness mapping from interactions to hardness and iterating between hardness evaluation and CF recommender refinement.

Benefiting from this adversarial framework, our AdvInfoNCE endows two appealing properties. On the one hand, it serves as a specialized contrastive loss for top-$K$ collaborative filtering that acknowledges the risk of false negatives while utilizing hard negative mining. Additionally, we theoretically demonstrate through adaptive gradients that AdvInfoNCE subtly employs informative negative sampling (*cf.* Appendix B.3). On the other hand, we bridge the adversarial hardness learned by AdvInfoNCE with the ambiguity set in distributionally robust optimization (DRO), thereby naturally demonstrating its generalization ability and assuring its robust performance against the noise of false negatives (*cf.* Theorem 3.1). This furnishes a potential theoretical rationalization for the exceptional robustness observed in contrastive learning. Extensive experiments on the out-of-distribution benchmark datasets in top-$K$ recommendation tasks further highlight the capability of AdvInfoNCE in addressing the issue of biased observations. We advocate for AdvInfoNCE to be considered as a significant reference loss for future contrastive learning research in CF and recommend its adoption as a standard loss in recommender systems.

## 2 Preliminary of Contrastive Collaborative Filtering (CF)

**Task Formulation.** Personalized recommendations aim to retrieve a small subset of items from a large catalog to align with the user preference. Here we focus on a typical setting, collaborative filtering (CF) with implicit feedback (*e.g.,* click, purchase, view times, etc.), which can be framed as a semi-supervised top-$K$ recommendation problem [29]. Wherein, the majority of all possible user-item interactions are unobserved, thus previous studies typically assign them with negative labels [29, 30]. Let $\mathcal{O}^+ = \{(u,i)|y_{ui} = 1\}$ ($\mathcal{O}^- = \{(u,i)|y_{ui} = 0\}$) be the observed (unobserved) interactions between users $\mathcal{U}$ and items $\mathcal{I}$, where $y_{ui} = 1$ ($y_{ui} = 0$) indicates that user $u \in \mathcal{U}$ has (has not) interacted with item $i \in \mathcal{I}$. For convenience, let $\mathcal{I}_u^+ = \{i|y_{ui} = 1\}$ ($\mathcal{I}_u^- = \{j|y_{uj} = 0\}$) denote the set of items that user $u$ has (has not) adopted before. To overcome the distribution shifts in real-world scenarios [31, 10], the long-acting goal is to optimize a robust CF model $\hat{y} : \mathbb{U} \times \mathbb{I} \to \mathbb{R}$ capable of distilling the true preferences of users towards items.

**Modeling Strategy.** Leading CF models [32–35] involve three main modules: a user behavior encoder $\psi_\theta(\cdot) : \mathbb{U} \to \mathbb{R}^d$, an item encoder $\phi_\theta(\cdot) : \mathbb{I} \to \mathbb{R}^d$, and a predefined similarity function $s(\cdot, \cdot) : \mathbb{R}^d \times \mathbb{R}^d \to \mathbb{R}$, where $\theta$ is the set of all trainable parameters. The encoders transfer the user and item into $d$-dimensional representations. The similarity function measures the similarity between the user and item in the representation space, whose widely-used implementations include dot product [30], cosine similarity [36], and neural networks [37]. For the sake of simplicity and better interpretation, we set the similarity function in our paper as:

$$s(u,i) = \frac{1}{\tau} \cdot \frac{\psi_\theta(u)^\top \phi_\theta(i)}{\|\psi_\theta(u)\| \cdot \|\phi_\theta(i)\|}, \tag{1}$$

in which $\tau$ is the hyper-parameter known as temperature [38].

**Loss Function.** Point- and pair-wise loss functions are widely used to optimize the parameters $\theta$:

- Point-wise losses (*e.g.,* binary cross-entropy [39, 40], mean square error [32]) typically treat observed interactions $\mathcal{O}^+$ as positive instances and all unobserved interactions $\mathcal{O}^-$ as negatives. CF methods equipped with point-wise loss naturally cast the problem of item recommendations into a binary classification or regression task. However, due to the large scale of indiscriminative unobserved interactions, this type of loss function fails to effectively consider the ranking criterion nor efficiently handle false negative instances [41].

- Pairwise loss functions (*e.g.,* BPR [30], WARP [42], pairwise hinge loss [43]) aim to differentiate items in a specific relative order. For instance, BPR assumes that the user prefers the positive item over unobserved items. Although effective, these loss functions share a common limitation that lacks sensitivity to label noise and false negatives [22].

Recent CF methods, inspired by the success of contrastive learning (CL) in self-supervised learning tasks, show a surge of interest in contrastive loss:

- Contrastive losses (*e.g.,* InfoNCE [16], InfoL1O [44], DCL [45], SupCon [46]) enforce the agreement between positive instances and the discrepancy between negative instances in the representation space. Separately treating the observed and unobserved interactions as positive and negative

instances allows us to get rid of data augmentation and directly apply these losses in CF. Here we focus mainly on InfoNCE (also well-known as softmax loss in CF [29]), and our findings could be easily extended to other contrastive losses. Specifically, given a user $u$ with a positive item $i$ and the set of items $\mathcal{I}_u^-$ that $u$ has not interacted with, InfoNCE essentially encourages the CF model to satisfy the following ranking criterion:

$$\forall j \in \mathcal{I}_u^-, \quad i >_u j, \tag{2}$$

where $>_u$ represents the personalized ranking order of user $u$. We equivalently reformulate this ranking criterion in the semantic similarity space, as depicted in Figure 1a:

$$\forall j \in \mathcal{I}_u^-, \quad s(u, j) - s(u, i) \leq 0. \tag{3}$$

Overall, the objective function of personalized recommendation with InfoNCE loss can be formulated as follows:

$$\mathcal{L}_{\text{InfoNCE}} = - \sum_{(u,i) \in \mathcal{O}^+} \log \frac{\exp\left(s(u, i)\right)}{\exp\left(s(u, i)\right) + \sum_{j \in \mathcal{N}_u} \exp\left(s(u, j)\right)}, \tag{4}$$

where $(u, i) \in \mathcal{O}^+$ is an observed interaction between user $u$ and item $i$. In most cases, the number of unobserved items, *i.e.*, $|\mathcal{I}_u^-|$, can be extremely large, thus negative sampling becomes a prevalent solution. The standard process of negative sampling involves uniformly drawing a subset of negative instances, denoted by $\mathcal{N}_u$, from the unobserved item set $\mathcal{I}_u^-$.

**Discussion.** Scrutinizing the ranking criterion in Equation (3), we can easily find that InfoNCE cares about the positive item's relative orders with the negative items holistically, while ignoring the ranking relationships among these negatives. Moreover, it simply treats all unobserved interactions as negative items, thus easily overlooking the risk of false negatives [29, 30]. Such oversights impede the performance and generalization ability of InfoNCE-empowered CF models. To bridge the gap, we aim to devise a variant of InfoNCE by considering these factors.

## 3 Methodology of AdvInfoNCE

On the foundation of InfoNCE loss, we first devise AdvInfoNCE, a robust contrastive loss tailor-made for Top-$K$ recommendation scenarios. Then we present its desirable properties that underscore the efficacy of negative sampling and generalization ability across distribution shifts. Additionally, we delve into the hard negative mining mechanism of AdvInfoNCE and its alignment with Top-$K$ ranking evaluation metrics in Appendix B.3 and B.4, respectively.

### 3.1 Fine-grained Ranking Criterion

Instead of relying on the coarse relative ranking criterion presented in Equation (3), we learn a fine-grained ranking criterion in Top-$K$ recommendation by incorporating varying degrees of user preferences for different items into the loss function. In other words, we need to treat the positive item as an anchor, quantify the relative hardness of each item as compared with the anchor, and differentiate between false negatives and hard negatives. Here the hardness of a negative item is defined as its minimal similarity distance towards the positive anchor.

For simplicity, we consider a single observed user-item pair $(u, i) \in \mathcal{O}^+$ from now on. Let us assume that for $(u, i)$, we have access to the oracle hardness $\delta_j$ of each negative item $j \in \mathcal{N}_u$. By assigning distinct hardness scores to different negatives, we reframe a hardness-aware fine-grained ranking criterion in the semantic similarity space as follows:

$$\forall j \in \mathcal{N}_u, \quad s(u, j) - s(u, i) + \delta_j < 0. \tag{5}$$

Incorporating the hardness scores enables us to exhibit the ranking among negative items (*i.e.,* the larger hardness score $\delta_j$ the item $j$ holds, the less likely the user $u$ would adopt it) and endow with better flexibility, compared with InfoNCE in Figure 1a. Specifically, as Figure 1b shows, if $\delta_{j_1} > 0$, $j_1$ is identified as a hard negative item with a contracted feasible zone, which indicates a stringent constraint prompting the model to focus more on $j_1$. Meanwhile, as Figure 1c depicts, if $\delta_{j_2} < 0$, $j_2$ tends to be a false negative item and its feasible zone expands, which characterizes a more relaxed constraint. Clearly, the hardness degree $\delta_j$ encapsulates a fine-grained ranking criterion for user $u$, thereby facilitating more accurate recommendation.

## 3.2 Derivation of AdvInfoNCE

The fine-grained ranking criterion for a single positive interaction $(u, i)$, as defined in Equation (5), can be seamlessly transformed into the main part of our AdvInfoNCE:

$$\underbrace{\max\{0, \{s(u, j) - s(u, i) + \delta_j\}_{j \in \mathcal{N}_u}\}}_{\text{Fine-grained ranking criterion}} \approx \underbrace{-\log\{\frac{\exp(s(u, i))}{\exp(s(u, i)) + \sum_{j=1}^{|\mathcal{N}_u|} \exp(\delta_j)\exp(s(u, j))}\}}_{\text{AdvInfoNCE}} \quad (6)$$

InfoNCE is a special case of AdvInfoNCE, when $\forall j, \delta_j = 0$. We briefly outline the main steps of derivation (see Appendix B.1 for the complete derivation):

*Derivation Outline.* Based on the LogSumExp operator, *i.e.,* $\max(x_1, ..., x_n) \approx \log(\sum_i \exp(x_i))$, the fine-grained ranking criterion could be approximated in a different form as $\log(\exp(0) + \sum_{j=1}^{|\mathcal{N}_u|} \exp(s(u, j) - s(u, i) + \delta_j))$. Consequently, the right-hand side of Equation (6) can be derived by factoring out the negative sign of the logarithm and reorganizing the term. $\square$

We would like to highlight the straightforward nature of this derivation, which is achieved without the need for assumptions, but the widely-used LogSumExp operator [47].

However, the oracle hardness $\delta_j$ is not accessible or measurable in real-world scenarios. The challenge, hence, comes to how to automatically learn the appropriate hardness for each negative instance. Inspired by the adversarial training [48], we exploit a min-max game, which allows to train the model alternatively between the prediction of hardness and the refinement of CF model. Formally, taking into account all observed interactions, we cast the AdvInfoNCE learning framework as the following optimization problem:

$$\min_{\theta} \mathcal{L}_{\text{AdvInfoNCE}} = \min_{\theta} \max_{\Delta \in \mathbb{C}(\eta)} - \sum_{(u,i) \in \mathcal{O}^+} \log \frac{\exp(s(u, i))}{\exp(s(u, i)) + \sum_{j=1}^{|\mathcal{N}_u|} \exp(\delta_j^{(u,i)})\exp(s(u, j))} \quad (7)$$

Here, with a slight abuse of notation, let $\delta_j^{(u,i)}$ denote the relative hardness of negative item $j$ in contrast with interaction $(u, i)$, and $\Delta$ represent the collective set of all $\delta_j^{(u,i)}$. The definition and enlightening explanation of $\mathbb{C}(\eta)$ will be presented in Section 3.3.

## 3.3 In-depth Analysis of AdvInfoNCE

In spite of the adversarial training framework of AdvInfoNCE, the theoretical understanding in terms of its inherent generalization ability remains untouched. In this section, we study it through the lens of distributionally robust optimization (DRO) [49] over negative sampling. From this perspective, AdvInfoNCE focuses on the worst-case distribution over high-quality hard negative sampling.

**Theorem 3.1.** *We define $\delta_j^{(u,i)} \doteq \log(|\mathcal{N}_u| \cdot p(j|(u, i)))$, where $p(j|(u, i))$ is the probability of sampling negative item $j$ for observed interaction $(u, i)$. Then, optimizing AdvInfoNCE loss is **equivalent** to solving Kullback-Leibler (KL) divergence-constrained distributionally robust optimization (DRO) problems over negative sampling:*

$$\min_{\theta} \mathcal{L}_{AdvInfoNCE} \iff \min_{\theta} \max_{p(j|(u,i)) \in \mathbb{P}} \mathbb{E}_P[\exp(s(u, j) - s(u, i)) : \mathcal{D}_{KL}(P_0 || P) \leq \eta] \quad (8)$$

*where $P_0$ stands for the distribution of uniformly drawn negative samples,* i.e., *$p_0(j|(u, i)) = \frac{1}{|\mathcal{N}_u|}$; $P$ denotes the distribution of negative sampling $p(j|(u, i))$.*

For brevity, we present a proof sketch summarizing the key idea here and delegate the full proof to Appendix B.2.

*Proof.* By defining $\delta_j^{(u,i)} \doteq \log(|\mathcal{N}_u| \cdot p(j|(u, i)))$, we have $\frac{1}{|\mathcal{N}_u|} \sum_{j=1}^{|\mathcal{N}_u|} \delta_l^{(u,i)} = -\mathcal{D}_{KL}(P_0 || P)$. Therefore, by rearranging the terms, optimizing AdvInfoNCE can be restated as a standard DRO

problem:

$$\min_{\theta} \mathcal{L}_{\text{AdvInfoNCE}} = \min_{\theta} \max_{\Delta \in \mathcal{C}(\eta)} \sum_{(u,i) \in \mathcal{O}^+} \log(1 + |\mathcal{N}_u| \sum_{j=1}^{|\mathcal{N}_u|} p(j|(u,i)) \exp(s(u,j) - s(u,i)))$$

$$\iff \min_{\theta} \sum_{(u,i) \in \mathcal{O}^+} \sup_{p(j|(u,i)) \in \mathbb{P}} \mathbb{E}_P[\exp(s(u,j) - s(u,i))], \text{ s.t. } \mathcal{D}_{KL}(P_0||P) \leq \eta$$

$\square$

We now turn our attention to the role of $\mathcal{C}(\eta)$ and its relationship with the ambiguity set $\mathbb{P}$ in DRO. The ambiguity set $\mathbb{P}$ is formulated by requiring the distribution of negative sampling to fall within a certain $\eta$ distance from the uniform distribution, as defined below:

$$\mathbb{P} = \{p(j|(u,i)) \in (\frac{1}{|\mathcal{N}_u|} - \epsilon, \frac{1}{|\mathcal{N}_u|} + \epsilon) : \mathcal{D}_{KL}(P_0||P) \leq \eta = -log(1 - \frac{\epsilon^2}{|\mathcal{N}_u|})\} \quad (9)$$

where $\epsilon$ serves as the hyperparameter to regulate the deviation of the negative sampling distribution from the uniform distribution. In terms of implementation, $\epsilon$ is controlled by the number of adversarial training epochs (see Algorithm 1). Clearly, $\mathcal{C}(\eta)$ is an equal counterpart to the ambiguity set $\mathbb{P}$.

**Discussion.** Grounded by theoretical proof, we understand the generalization ability of our AdvInfoNCE through the lens of DRO over informative negative sampling. In essence, by learning the hardness in an adversarial manner, AdvInfoNCE effectively enhances the robustness of the CF recommender. Moreover, apart from the robustness of AdvInfoNCE (see Section 3.3), we also identify its hard negative mining mechanism through gradients analysis (see Appendix B.3) and its alignment with top-$K$ ranking evaluation metrics (see Appendix B.4). Specifically, the gradients concerning negative samples are proportional to the hardness term $\exp(\delta_j)$, indicating that AdvInfoNCE is a hardness-aware loss. To some extent, our AdvInfoNCE is equivalent to the widely-adopted Discounted Cumulative Grain (DCG) ranking metric.

**Limitation.** AdvInfoNCE employs end-to-end negative sampling in an adversarial manner, compelling the recommender to ensure robustness from the worst-case scenario. Despite its empirical success and desirable properties, AdvInfoNCE is not immune to the limitations of adversarial training, which is well-known for its potential training instability.

## 4 Experiments

We aim to answer the following research questions:

- **RQ1:** How does AdvInfoNCE perform compared with other CL-based CF methods?
- **RQ2:** Can AdvInfoNCE effectively learn the fine-grained ranking criterion? What are the impacts of the component $\epsilon$ (*i.e.,* the number of adversarial training epochs) on AdvInfoNCE?

**Baselines.** Two high-performing encoders - ID-based (MF [50]) and graph-based (LightGCN [51]), are selected as CF backbone models. We thoroughly compare AdvInfoNCE with two categories of the latest CL-based CF methods: augmentation-based baselines (SGL [13], NCL [52], XSimGCL [27]) and loss-based baselines (CCL [53], BC Loss [54], Adap-$\tau$ [55]). See detailed introduction and comparison of baselines in Appendix A.

**Datasets.** To verify the generalization ability of AdvInfoNCE, we conduct extensive experiments on three standard unbiased datasets (KuaiRec [56], Yahoo!R3 [57], Coat [58]) and one synthetic dataset (Tencent [10]). We employ three commonly used metrics (Hit Ratio (HR@$K$), Recall@$K$, Normalized Discounted Cumulative Gain (NDCG@$K$)) for evaluation, with $K$ set by default at 20. Please refer to Appendix C and D for more experimental results over additional backbones, additional baselines, and an intuitive understanding of AdvInfoNCE.

### 4.1 Overall Performance Comparison (RQ1)

#### 4.1.1 Evaluations on Unbiased Datasets

**Motivation.** Evaluating the efficacy and generalization ability of CF models based on partially observed interactions collected from existing recommender system is rather challenging [59]. This is

Table 1: The performance comparison on unbiased datasets over the LightGCN backbone. The improvement achieved by AdvInfoNCE is significant ($p$-value $<< 0.05$).

| | KuaiRec | | Yahoo!R3 | | Coat | |
|---|---|---|---|---|---|---|
| | Recall | NDCG | Recall | NDCG | Recall | NDCG |
| BPR (Rendle et al., 2012) | 0.1652 | 0.3905 | 0.1487 | 0.0697 | 0.2737 | 0.1707 |
| InfoNCE (van den Oord et al., 2018) | 0.1800 | 0.4529 | 0.1475 | 0.0698 | 0.2689 | 0.1882 |
| SGL (Wu et al., 2021) | $0.1829^{+1.61\%}$ | $\underline{0.4583}^{+1.19\%}$ | $0.1474^{-0.07\%}$ | $0.0692^{-0.86\%}$ | $0.2737^{+1.79\%}$ | $0.1716^{-8.82\%}$ |
| NCL (Lin et al., 2022) | $0.1764^{-2.00\%}$ | $0.4478^{-1.13\%}$ | $0.1407^{-4.61\%}$ | $0.0669^{-4.15\%}$ | $\underline{0.2778}^{+3.31\%}$ | $0.1739^{-7.60\%}$ |
| XSimGCL (Yu et al., 2022) | $\underline{0.1907}^{+5.94\%}$ | $0.4531^{+0.04\%}$ | $0.1503^{+1.90\%}$ | $\underline{0.0708}^{+1.43\%}$ | $0.2729^{+1.49\%}$ | $0.1655^{-12.06\%}$ |
| CCL (Mao et al., 2021) | $0.1776^{-1.33\%}$ | $0.4497^{-0.71\%}$ | $0.1453^{-1.49\%}$ | $0.0676^{-3.15\%}$ | $0.2732^{+1.60\%}$ | $0.1941^{+3.13\%}$ |
| BC Loss (Zhang et al., 2022) | $0.1799^{-0.06\%}$ | $0.4417^{-2.47\%}$ | $0.1498^{+1.56\%}$ | $0.0703^{+0.72\%}$ | $0.2719^{+1.12\%}$ | $0.1921^{+2.07\%}$ |
| Adap-$\tau$ (Chen et al., 2023) | $0.1717^{-4.61\%}$ | $0.4323^{-4.55\%}$ | $\underline{0.1516}^{+2.78\%}$ | $0.0704^{+0.86\%}$ | $0.2700^{+0.41\%}$ | $\underline{0.1957}^{+3.99\%}$ |
| **AdvInfoNCE** | $\mathbf{0.1979}*^{+9.94\%}$ | $\mathbf{0.4697}*^{+3.71\%}$ | $\mathbf{0.1527}*^{+3.53\%}$ | $\mathbf{0.0718}*^{+2.87\%}$ | $\mathbf{0.2846}*^{+5.84\%}$ | $\mathbf{0.2026}*^{+7.65\%}$ |

primarily due to the exposure bias of the recommender system confounding the observed data. To address this issue, some researchers propose using unbiased evaluations [57, 58]. In these datasets, items in the test set are randomly exposed to users, while partially observed interactions are still used for training. The distribution shift between training and testing enables a fair evaluation of the generalization ability of CF methods in out-of-distribution tasks.

**Results.** Tables 1 and 4 present a comparative analysis of the performance on standard unbiased datasets over the LightGCN and MF backbones. The best-performing methods per test are bold and starred, while the second-best methods are underlined. The red and blue percentages respectively refer to the increase and decrease of CF methods relative to InfoNCE in each metric. We observe that:

- **CL-based CF methods have driven impressive performance breakthroughs compared with BPR in most evaluation metrics.** This aligns with findings from prior research [27, 54] that contrastive loss significantly boosts the generalization ability of CF methods by providing informative contrastive signals. In contrast, BPR loss, which focuses on pairwise relative ranking, often suffers from ineffective negative sampling [30] and positive-unlabeled issues [21].

- **Contrastive loss, rather than graph augmentation, plays a more significant role in enhancing the generalization ability of CF models.** We categorize CL-based CF baselines into two research lines: one adopts user-item bipartite graph augmentation along with augmented views as positive signals (referred to as augmentation-based), and another focuses solely on modifying contrastive loss using interacted items as positive instances (referred to as loss-based). Surprisingly, our results reveal no substantial difference between the performance of augmentation- and loss-based methods. This finding provides further evidence in support of the claim made in prior studies [27, 26] that, specifically in CF tasks, the contrastive loss itself is the predominant factor contributing to the enhancement of a CF model's generalization ability.

- **AdvInfoNCE consistently outperforms the state-of-the-art CL-based CF baselines in terms of all metrics on all unbiased datasets.** AdvInfoNCE shows an improvement ranging from 3.53% to 9.94% on Recall@20 compared to InfoNCE. Notably, AdvInfoNCE gains the most on the fully-exposed dataset, KuaiRec, which is considered as an ideal offline A/B testing. In contrast, most of the CL-based CF methods underperform relative to InfoNCE and behave in an unstable manner when the testing distribution shifts. This clearly indicates that AdvInfoNCE greatly enhances the robustness of the CF model in real-world scenarios. We attribute this improvement to the fine-grained ranking criterion of AdvInfoNCE that automatically differentiates hard and false negative samples. Additionally, as shown in Table 9, AdvInfoNCE only adds negligible time complexity compared to InfoNCE.

### 4.1.2 Evaluations on Various Out-of-distribution Settings

**Motivation.** In real-world scenarios, recommender systems may confront diverse, unpredictable, and unknown distribution shifts. We believe that a good recommender with powerful generalization ability should be able to handle various degrees of distribution shift. Following previous work [10], we partition the Tencent dataset into three test sets with different out-of-distribution degrees. Here, a higher $\gamma$ value signifies a greater divergence in distribution shift (for more details on the experimental settings, see Appendix C.1). We evaluate all CF methods with identical model parameters across all test sets, without knowing any prior knowledge of the test distribution beforehand.

Table 2: The performance comparison on the Tencent dataset over the LightGCN backbone. The improvement achieved by AdvInfoNCE is significant ($p$-value $<< 0.05$).

| | $\gamma = 200$ | | | $\gamma = 10$ | | | $\gamma = 2$ | | | Validation |
| | HR | Recall | NDCG | HR | Recall | NDCG | HR | Recall | NDCG | NDCG |
|---|---|---|---|---|---|---|---|---|---|---|
| BPR (Rendle et al., 2012) | 0.1141 | 0.0416 | 0.0233 | 0.0720 | 0.0262 | 0.0146 | 0.0501 | 0.0186 | 0.0109 | 0.0673 |
| InfoNCE (van den Oord et al., 2018) | 0.1486 | 0.0540 | 0.0320 | 0.0924 | 0.0332 | 0.0195 | 0.0646 | 0.0242 | 0.0145 | 0.0854 |
| SGL (Wu et al., 2021) | 0.1227 | 0.0455 | 0.0249 | 0.0756 | 0.0281 | 0.0157 | 0.0517 | 0.0198 | 0.0113 | 0.0729 |
| NCL (Lin et al., 2022) | 0.1132 | 0.0413 | 0.0226 | 0.0734 | 0.0274 | 0.0145 | 0.0511 | 0.0189 | 0.0123 | 0.0669 |
| XSimGCL (Yu et al., 2022) | 0.1392 | 0.0501 | 0.0273 | 0.0830 | 0.0297 | 0.0160 | 0.0539 | 0.0203 | 0.0110 | 0.0873 |
| CCL (Mao et al., 2021) | 0.1372 | 0.0516 | 0.0318 | 0.0925 | 0.0350 | 0.0221 | 0.0683 | 0.0266 | 0.0170 | 0.0782 |
| BC Loss (Zhang et al., 2022) | 0.1513 | 0.0562 | 0.0341 | 0.0984 | 0.0366 | 0.0216 | 0.0682 | 0.0260 | 0.0164 | 0.0817 |
| Adap-$\tau$ (Chen et al., 2023) | 0.1488 | 0.0537 | 0.0317 | 0.0940 | 0.0338 | 0.0200 | 0.0642 | 0.0239 | 0.0143 | 0.0852 |
| **AdvInfoNCE** | **0.1600*** | **0.0594*** | **0.0356*** | **0.1087*** | **0.0403*** | **0.0243*** | **0.0774*** | **0.0295*** | **0.0180*** | 0.0879 |
| Imp.% over the strongest baselines | 5.74% | 5.75% | 4.53% | 10.50% | 10.11% | 9.95% | 13.32% | 10.90% | 5.88% | – |
| Imp.% over InfoNCE | 7.67% | 10.00% | 11.25% | 17.64% | 21.39% | 24.62% | 19.81% | 21.90% | 24.14% | – |

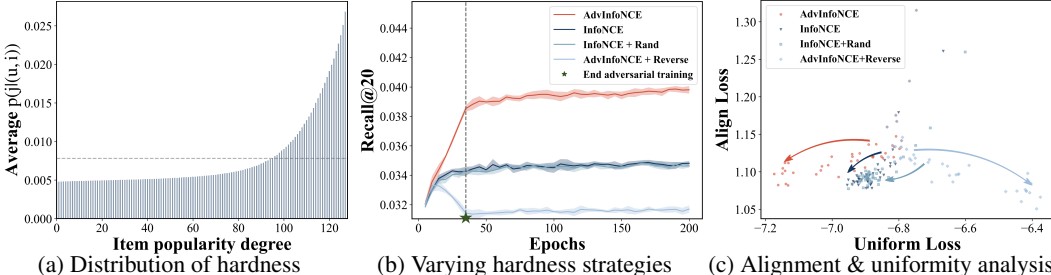

(a) Distribution of hardness   (b) Varying hardness strategies   (c) Alignment & uniformity analysis

Figure 2: Study of hardness. (2a) Illustration of hardness *i.e.,* the probability of negative sampling ($p(j|(u, i))$) learned by AdvInfoNCE *w.r.t.* item popularity on Tencent. The dashed line represents the uniform distribution. (2b) Performance comparisons with varying hardness learning strategies on Tencent ($\gamma = 10$). (2c) The trajectories of align loss and uniform loss during training progress. Lower values indicate better performance. Arrows denote the losses' changing directions.

**Results.** As illustrated in Tables 2 and 5, AdvInfoNCE yields a consistent boost compared to all SOTA baselines across all levels of out-of-distribution test sets. In particular, AdvInfoNCE substantially improves Recall@20 by 10.00% to 21.90% compared to InfoNCE and by 5.75% to 10.90% compared to the second-best baseline. It's worth noting that AdvInfoNCE gains greater improvement on test sets with higher distribution shifts, while all CL-based CF baselines suffer significant performance drops due to these severe distribution shifts. We attribute this drop to the coarse relative ranking criterion which injects erroneously recognized negative instances into contrastive signals. In contrast, benefiting from the effective adversarial learning of the fine-grained ranking criterion, AdvInfoNCE can achieve a 24.14% improvement over InfoNCE *w.r.t.* NDCG@20 when $\gamma = 2$, fully stimulating the potential of contrastive loss. These results validate the strong generalization ability of AdvInfoNCE as demonstrated in Theorem 3.1, proving that optimizing AdvInfoNCE is equivalent to solving DRO problems constrained by KL divergence over high-quality negative sampling.

## 4.2 Study on AdvInfoNCE (RQ2)

### 4.2.1 Effect of Hardness

**Motivation.** To evaluate the effectiveness of the fine-grained hardness-aware ranking criterion learned by AdvInfoNCE, and further investigate whether it truly enhances the generalization ability of CF models, we conduct a comprehensive analysis on the Tencent dataset. This analysis contains three-fold: it studies the distribution of hardness (*i.e.,* examining the negative sampling strategy of AdvInfoNCE); the performance trend over varying hardness learning strategies; and conducting an analysis of the alignment and uniformity properties of representations over varying hardness learning strategies. Additional experiments on the KuaiRec dataset, exhibiting similar trends and findings, can be found in Appendix D.2.

**Distribution of hardness, *aka.* negative sampling strategy.** As elucidated in Theorem 3.1, learning the hardness $\delta_j^{(u,i)}$ in an adversarial manner essentially conducts negative sampling. Inspired by item popularity-based negative sampling strategies [60–62], we probe into the relationship between adversarial negative sampling of AdvInfoNCE and the popularity of negative items. Figure 2a

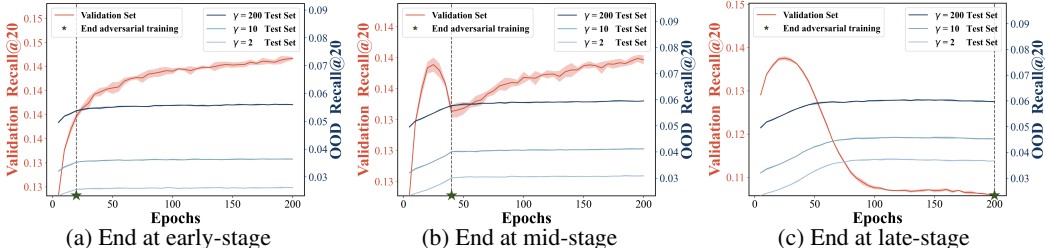

Figure 3: The performance on the in-distribution validation set and out-of-distribution test sets of Tencent. The total number of epochs of adversarial training is gradually increasing from (a) to (c), and the green star denotes the timepoint when we terminate adversarial training.

depicts a skewed negative sampling distribution that focuses more on popular negative items and de-emphasizes the hardness of unpopular negative items, which are likely to be false negatives. These results highlight two findings. Firstly, in line with [61], only a minor portion of negative instances are potentially important for model learning, while a vast number of negatives are false negatives. Secondly, more frequent items constitute better hard negatives, which is consistent with the findings in previous work [60]. This popularity-correlated negative sampling validates that AdvInfoNCE automatically mitigates the influence of popularity bias during training, alleviates the overlook of false negatives in CF, and utilizes high-quality hard negative mining.

**Performance over varying hardness learning strategies.** Figure 2b clearly demonstrates that distilling the fine-grained hardness of negative items (AdvInfoNCE) is excelling over the uniformly random hardness (InfoNCE+Rand), equal hardness (InfoNCE), and reversed fine-grained hardness (AdvInfoNCE+Reverse) throughout the training process. This validates AdvInfoNCE's ability to simultaneously filter and up-weight more informative hard negatives while identifying and down-weighting the potential false negatives. These desired properties further enable AdvInfoNCE to extract high-quality negative signals from a large amount of unobserved data.

**Alignment and uniformity analysis.** It is widely accepted that both alignment and uniformity are necessary for a good representation [5]. We, therefore, study the behavior of AdvInfoNCE through the lens of align loss and uniform loss, as shown in Figure 2c. AdvInfoNCE improves uniformity while maintaining a similar level of alignment as InfoNCE, whereas reversed fine-grained hardness significantly reduces representation uniformity. These findings underscore the importance of fine-grained hardness, suggesting that AdvInfoNCE learns more generalized representations.

### 4.2.2 Effect of the Adversarial Training Epochs

The critical hyperparameter, the number of adversarial training epochs, controls the degree of deviation from uniform negative sampling. As illustrated in Figure 3a through 3b, improved out-of-distribution performance correlates with an increased adversarial learning period. On the other hand, Figures 3b through 3c show that overly prolonged adversarial training can also degrade performance on in-distribution validation sets. Therefore, it is crucial to devise a proper adversarial training period tailored to specific datasets, which aligns with the limitation of AdvInfoNCE. Consistent observations can be found on KuaiRec in Appendix D.3.

## 5 Conclusion

Despite the empirical successes, the profound understanding of contrastive loss in collaborative filtering (CF) remains limited. In this work, we devised a principled AdvInfoNCE loss specially tailored for CF methods, utilizing a fine-grained hardness-aware ranking criterion to enhance the recommender's generalization ability. Grounded by theoretical proof, AdvInfoNCE empowers top-$K$ recommendations via informative negative sampling in an adversarial manner. We believe that our AdvInfoNCE, as a variant of InfoNCE, provides a valuable baseline for future contrastive learning-based CF studies and is an important stepping stone toward generalized recommenders.

## Acknowledgments and Disclosure of Funding

This research is supported by the NExT Research Center, the National Natural Science Foundation of China (9227010114), the University Synergy Innovation Program of Anhui Province (GXXT-2022-040), and the MOE Project of Key Research Institute of Humanities and Social Sciences (22JJD110001).

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

# A    Related Work

We remind important related works to understand how our AdvInfoNCE stands and its role in rich literature. Our work is related to the literature on contrastive learning-based collaborative filtering (CL-based CF) methods, and theoretical understanding of contrastive loss in collaborative filtering.

## A.1    Contrastive Learning-based Collaborative Filtering

The latest **CL-based CF methods** can roughly fall into two research lines. The first one, which we term the "augmentation-based" approach, leverages user-item bipartite graph augmentations along with augmented views as positive signals. The second category, referred to as "loss-based" approaches, mainly focuses on the modification of contrastive loss. In loss-based CF models, interacted items serve as positive instances.

- **Augmentation-based [13–15, 52, 26, 27, 63–66].** The prevailing augmentation-based paradigm in CL-based CF methods is to employ user-item bipartite graph augmentations to generate contrasting views. These contrasting views are then treated as positive instances in the application of contrastive loss, such as InfoNCE loss, to further enhance collaborative filtering signals. Recent studies have extensively explored methods for generating contrastive views. Several studies like SGL [13] and DCL [64] elaborate on data-heuristic augmentation operators such as a random node or edge dropout and random walk. NCL [52] takes a different approach and incorporates potential neighbors from both the graph structure and semantic space into contrastive views. XSimGCL [27] takes it a step further and discards ineffective graph augmentations, choosing instead to employ a simple noise-based embedding augmentation. In pursuit of high-quality augmented supervision signals instead of handcrafted strategies, AutoCF [15] designs a learnable masking function to automatically identify important centric nodes for data augmentation.
- **Loss-based [53–55, 22, 67].** Recent research, such as the experiments presented in SimGCL [26] and XSimGCL [27], has empirically shown that contrastive loss can be instrumental in enhancing the performance of CF methods, often playing a more significant role than heuristic-based graph augmentation. Despite these findings, there remains a gap in the exploration of loss-based CF methods, an area ripe for further investigation. BC loss [54] incorporates bias-aware margins into the contrastive loss, enabling the learning of high-quality head and tail representations with robust discrimination and generalization abilities. Adap-$\tau$ [55] proposes an adaptive fine-grained strategy for selecting the personalized temperature $\tau$ for each user within the contrastive loss. HDCCF [22] devises a new contrastive loss function extending the advantage of negative mining from user-item to neighbored users and items.

## A.2    Theoretical Understanding of Contrastive Loss in CF

Despite the remarkable success of CL-based CF methods, there remains a lack of theoretical understanding, particularly regarding the superior generalization ability of contrastive loss. In a study conducted by [68], three model-agnostic advantages of contrastive loss are theoretically revealed, including mitigating popularity bias, mining hard negative samples, and maximizing the ranking metric. CLRec [14] sheds light on contrastive loss from a bias-reduction perspective by revealing its connection with inverse propensity weighting techniques. XSimGCL [27] suggests that contrastive learning enables the recommender to learn more evenly distributed user and item representations, thereby mitigating the prevalent popularity bias in CF.

# B    In-depth Analysis of AdvInfoNCE

## B.1    Complete Derivation of AdvInfoNCE

*Full Derivation.* We first introduce the widely-used LogSumExp operator in machine learning algorithms.

$$\max(x_1, x_2, ..., x_n) \approx \log(\exp(x_1) + \exp(x_2) + ... + \exp(x_n)) \tag{10}$$

The fine-grained ranking criterion for a single positive interaction $(u, i)$ is defined as:

$$\forall j \in \mathcal{N}_u, \quad s(u, j) - s(u, i) + \delta_j < 0. \tag{11}$$

Then we probe into the Eq ([11]) and transform it into an optimization problem as follows:

$$\min_{\theta} \max\{0, \{s(u,j) - s(u,i) + \delta_j\}_{j \in \mathcal{N}_u}\} \tag{12}$$

We can seamlessly transform this optimization objective into the core component of our AdvInfoNCE:

$$\underbrace{\max\{0, \{s(u,j) - s(u,i) + \delta_j\}_{j \in \mathcal{N}_u}\}}_{\text{Hardness-aware ranking criterion}} \approx \log\left(\exp(0) + \sum_{j=1}^{|\mathcal{N}_u|} \exp(s(u,j) - s(u,i) + \delta_j)\right)$$

$$= \log\left\{1 + \sum_{j=1}^{|\mathcal{N}_u|} \exp(\delta_j)\exp(s(u,j) - s(u,i))\right\}$$

$$= \log\left\{1 + |\mathcal{N}_u| \sum_{j=1}^{|\mathcal{N}_u|} \frac{\exp(\delta_j)}{|\mathcal{N}_u|} \exp(s(u,j) - s(u,i))\right\}$$

$$= -\log\underbrace{\left\{\frac{\exp(s(u,i))}{\exp(s(u,i)) + |\mathcal{N}_u| \sum_{j=1}^{|\mathcal{N}_u|} \frac{\exp(\delta_j)}{|\mathcal{N}_u|} \exp(s(u,j))}\right\}}_{\text{AdvInfoNCE}}$$

$$\tag{13}$$

Drawing inspiration from adversarial training [48], we utilize a min-max game that allows for alternating training of the model between predicting hardness and refining the CF model. Formally, we formulate the AdvInfoNCE learning framework as the following optimization problem:

$$\min_{\theta} \mathcal{L}_{\text{AdvInfoNCE}} = \min_{\theta} \max_{\Delta \in \mathbb{C}(\eta)} - \sum_{(u,i) \in \mathcal{O}^+} \log \frac{\exp(s(u,i))}{\exp(s(u,i)) + |\mathcal{N}_u| \sum_{j=1}^{|\mathcal{N}_u|} \frac{\exp(\delta_j^{(u,i)})}{|\mathcal{N}_u|} \exp(s(u,j))} \tag{14}$$

where $\frac{\exp(\delta_j^{(u,i)})}{|\mathcal{N}_u|} \in \mathbb{C}(\eta, (u,i)) = (\frac{1}{|\mathcal{N}_u|} - \epsilon, \frac{1}{|\mathcal{N}_u|} + \epsilon)$, and $\epsilon$ is a hyperparameter that regulates the upper-bound deviation of hardness. In practice, $\epsilon$ is regulated by the number of adversarial training epochs under a fixed learning rate (refer to Algorithm [1]).

If we further define $\frac{\exp(\delta_j^{(u,i)})}{|\mathcal{N}_u|}$ as $p(j|(u,i))$, which signifies the likelihood of selecting item $j$ as a negative sample for the user-item pair $(u,i)$, we can rewrite the AdvInfoNCE loss as:

$$\min_{\theta} \mathcal{L}_{\text{AdvInfoNCE}} = \min_{\theta} \max_{p(\cdot|\cdot)} - \sum_{(u,i) \in \mathcal{O}^+} \log \frac{\exp(s(u,i))}{\exp(s(u,i)) + |\mathcal{N}_u| \sum_{j=1}^{|\mathcal{N}_u|} p(j|(u,i))\exp(s(u,j))} \tag{15}$$

$\square$

## B.2  Proof of Theorem

**Theorem 3.1.** *We define $\delta_j^{(u,i)} \doteq \log(|\mathcal{N}_u| \cdot p(j|(u,i)))$, where $p(j|(u,i))$ is the probability of sampling negative item $j$ for observed interaction $(u,i)$. Then, optimizing AdvInfoNCE loss is **equivalent** to solving Kullback-Leibler (KL) divergence-constrained distributionally robust optimization (DRO) problems over negative sampling:*

$$\min_{\theta} \mathcal{L}_{AdvInfoNCE} \iff \min_{\theta} \max_{p(j|(u,i)) \in \mathbb{P}} \mathbb{E}_P[\exp(s(u,j) - s(u,i)) : \mathcal{D}_{KL}(P_0||P) \leq \eta] \tag{8}$$

*where $P_0$ stands for the distribution of uniformly drawn negative samples, i.e., $p_0(j|(u,i)) = \frac{1}{|\mathcal{N}_u|}$; $P$ denotes the distribution of negative sampling $p(j|(u,i))$.*

*Proof.* We denote the relative hardness of negative item $j$ with respect to observed interaction $(u,i)$ as $\delta_j^{(u,i)}$ and redefine it as $\log(|\mathcal{N}_u| \cdot p(j|(u,i)))$. In this definition, $|\mathcal{N}u|$ is the number of negative

samples for each user $u$, and $p(j|(u, i))$ is the probability of selecting item $j$ as a negative sample for a given user-item pair $(u, i)$.

With the constraint that $\sum_{j=1}^{|\mathcal{N}_u|} p(j|(u, i)) = 1$, we can recalculate the average hardness as follows:

$$
\begin{aligned}
\frac{1}{|\mathcal{N}_u|} \sum_{j=1}^{|\mathcal{N}_u|} \delta_j^{(u,i)} &= \frac{1}{|\mathcal{N}_u|} \sum_{j=1}^{|\mathcal{N}_u|} \log(|\mathcal{N}_u| \cdot p(j|(u, i))) \\
&= - \sum_{j=1}^{|\mathcal{N}_u|} \frac{1}{|\mathcal{N}_u|} \log(\frac{1}{|\mathcal{N}_u|} \cdot \frac{1}{p(j|(u, i))}) \\
&= -\mathcal{D}_{KL}(P_0 || P)
\end{aligned}
\tag{16}
$$

In this formulation, $P_0$ represents the distribution of uniformly drawn negative samples, where $p_0(j|(u, i)) = \frac{1}{|\mathcal{N}_u|}$. Meanwhile, $P$ denotes the distribution of negative sampling, represented as $p(j|(u, i))$.

Since $p(j|(u, i)) \in \mathbb{C}(\eta, (u, i)) = (\frac{1}{|\mathcal{N}_u|} - \epsilon, \frac{1}{|\mathcal{N}_u|} + \epsilon)$, we further define $\eta = -log(1 - \frac{\epsilon^2}{|\mathcal{N}_u|})$. Thus, the feasible zone presented above is equivalent to a relaxed constraint $\mathcal{D}_{KL}(P_0 || P) \leq \eta$.

The AdvInfoNCE loss for all observations can be rewritten in a different form by reorganizing and simplifying the Eq (15):

$$
\begin{aligned}
\min_{\theta} \mathcal{L}_{\text{AdvInfoNCE}} = \min_{\theta} \sum_{(u,i) \in \mathcal{O}^+} \max_{p(j|(u,i)) \in \mathbb{P}} \log\{1 + |\mathcal{N}_u| \sum_{j=1}^{|\mathcal{N}_u|} p(j|(u, i)) \exp(s(u, j) - s(u, i))\} \\
\Longleftrightarrow \min_{\theta} \sum_{(u,i) \in \mathcal{O}^+} \max_{p(j|(u,i)) \in \mathbb{P}} \sum_{j=1}^{|\mathcal{N}_u|} p(j|(u, i)) \exp(s(u, j) - s(u, i)) \\
\Longleftrightarrow \min_{\theta} \sum_{(u,i) \in \mathcal{O}^+} \sup_{p(j|(u,i)) \in \mathbb{P}} \mathbb{E}_P[\exp(s(u, j) - s(u, i))]
\end{aligned}
\tag{17}
$$

The equation presented above exemplifies a widely encountered formulation of the Distributionally Robust Optimization (DRO) problem [49] where the ambiguity set of the probability distribution is defined by the Kullback-Leibler (KL) divergence $\mathcal{D}_{KL}(P_0 || P) \leq \eta$. $\qquad \square$

## B.3 Gradients Analysis

In this section, we delve into the crucial role of hardness $\delta_j$ in controlling the penalty strength on hard negative samples. The analysis begins with the main part of AdvInfoNCE for a single positive interaction $(u, i)$, as defined in Eq (13), primarily due to its simplicity. For the sake of notation simplicity, let us denote it as:

$$
\mathcal{L}_{\text{Adv}}(u, i) = -\log\{\frac{\exp(s(u, i))}{\exp(s(u, i)) + \sum_{j=1}^{|\mathcal{N}_u|} \exp(\delta_j) \exp(s(u, j))}\}
\tag{18}
$$

Then the gradient with respect to the positive representations $\phi_\theta(i)$ of item $i$ is formulated as:

$$
\begin{aligned}
-\nabla_i \mathcal{L}_{\text{Adv}}(u, i) &= \frac{\partial \mathcal{L}_{\text{Adv}}(u, i)}{\partial s(u, i)} \cdot \frac{\partial s(u, i)}{\partial \phi_\theta(i)} \\
&= (1 - \frac{\exp(s(u, i))}{\exp(s(u, i)) + \sum_{j=1}^{|\mathcal{N}_u|} \exp(\delta_j) \exp(s(u, j))}) \cdot \frac{\phi_\theta(i)}{\tau}
\end{aligned}
\tag{19}
$$

The gradients with respect to the negative representations $\phi_\theta(j)$ of item $j$ is given by:

$$-\nabla_j \mathcal{L}_{\text{Adv}}(u,i) = \frac{\partial \mathcal{L}_{\text{Adv}}(u,i)}{\partial s(u,j)} \cdot \frac{\partial s(u,j)}{\partial \phi_\theta(j)}$$

$$= \frac{\exp(\delta_j)\exp(s(u,j))}{\exp(s(u,i)) + \sum_{j=1}^{|\mathcal{N}_u|}\exp(\delta_j)\exp(s(u,j))} \cdot \frac{\phi_\theta(j)}{\tau}$$

$$= \exp(\delta_j)\{1 - \frac{\exp(s(u,i))}{\exp(s(u,i)) + \sum_{j=1}^{|\mathcal{N}_u|}\exp(\delta_j)\exp(s(u,j))}\}\frac{\exp(s(u,j))}{\sum_{j=1}^{|\mathcal{N}_u|}\exp(\delta_j)\exp(s(u,j))}\frac{\phi_\theta(j)}{\tau}$$

$$(20)$$

Clearly, for a given user $u$, the gradient with respect to the positive item $i$ equals the sum of gradients of all negative items, in accordance with the findings in [38]. The hardness $\exp(\delta_j)$ dictates the importance of negative gradients. Specifically, the gradients relating to the negative item $j$ in Eq (20) correlate proportionally to the hardness term $\exp(\delta_j)$, which shows that the AdvInfoNCE loss function is hardness-aware.

## B.4 Align Top-K evaluation metric

Discounted Cumulative Gain (DCG) is a commonly used ranking metric in top-$K$ recommendation tasks. In DCG, the relevance of an item's contribution to the utility decreases logarithmically in relation to its position in the ranked list. This mimics the behavior of a user who is less likely to scrutinize items that are positioned lower in the ranking. Formally, DCG over rank $\pi_s(u,i)$ is defined as follows:

$$DCG(\pi_s(u,\mathcal{I}),\mathbf{y}) = \sum_{i=1}^{|\mathcal{I}|}\frac{2^{y_i}-1}{\log_2(1+\pi_s(u,i))} \tag{21}$$

Where $\pi_s(u,\mathcal{I})$ is a ranked list over $\mathcal{I}$, as determined by the similarity function $s$ for user $u$, and $\mathbf{y}$ is a label vector that indicates whether an interaction has occurred previously or not. Then $\pi_s(u,i)$ is the rank of item $i$. Building on the research presented in [68], we explore how well AdvInfoNCE aligns with DCG for our purposes.

Under our proposed fine-grained hardness ranking criteria defined in Eq (11), the $\pi_s(u,i)$ can be obtained as follows:

$$\pi_s(u,i) = 1 + \sum_{j\in\mathcal{I}\setminus\{i\}}\mathbb{1}(s(u,j)-s(u,i)+\delta_j > 0)$$

$$\leq 1 + \sum_{j\in\mathcal{I}\setminus\{i\}}\exp(s(u,j)-s(u,i)+\delta_j). \tag{22}$$

The last inequality is satisfied by $\mathbb{1}(x>0) \leq \exp(x)$.

$$-\log[DCG(\pi_s(u,\mathcal{I}),\mathbf{y})] = -\log\left[\sum_{i=1}^{|\mathcal{I}|}\frac{2^{y_i}-1}{\log_2(1+\pi_s(u,i))}\right]$$

$$\leq -\log\left[\frac{1}{\log_2(1+\pi_s(u,i))}\right] \leq -\log\left[\frac{1}{\pi_s(u,i)}\right]$$

$$\leq -\log(\frac{1}{1+\sum_{j\in\mathcal{I}\setminus\{i\}}\exp(s(u,j)-s(u,i)+\delta_j)})$$

$$= -\log(\frac{\exp(s(u,i))}{\exp(s(u,i))+\sum_{j\in\mathcal{I}\setminus\{i\}}\exp(s(u,j)+\delta_j)})$$

$$= \mathcal{L}_{\text{Adv}}(u,i) \leq \mathcal{L}_{\text{AdvInfoNCE}} \tag{23}$$

Suppose that there are $K$ items in $\mathcal{I}$ that are interacted with $u$, let them to be $\{1,2,\cdots,K\}$ ithout loss of generality. Then

$$DCG(\pi_s(u,\mathcal{I}),\mathbf{y}) \leq \sum_{i=1}^{K}\frac{1}{\log_2(1+i)}, \tag{24}$$

the equality holds if and only if the interactions $\{(u, i) : i = 1, 2, \cdots, K\}$ are ranked top-K. For a given $u$,

$$\sum_{i=1}^{K} \mathcal{L}_{\text{Adv}}(u, i) = \sum_{i=1}^{K} - \log\left(\frac{\exp(s(u, i))}{\exp(s(u, i)) + \sum_{j \in \mathcal{I} \setminus \{i\}} \exp(s(u, j) + \delta_j)}\right)$$

$$= \sum_{i=1}^{K} \log\left(1 + \sum_{j \in \mathcal{I} \setminus \{i\}} \exp(s(u, j) - \exp(s(u, i)) + \delta_j)\right)$$

$$\geq \sum_{i=1}^{K} \log\left(1 + \sum_{j \in \mathcal{I} \setminus \{i\}} e^{-1}\right) = \sum_{i=1}^{K} \log(1 + (|\mathcal{I}| - 1)e^{-1}) \tag{25}$$

$$\geq \sum_{i=1}^{K} \frac{1}{\log_2(1 + i)} \geq DCG(\pi_s(u, \mathcal{I}), \mathbf{y}). \tag{26}$$

The inequality in Eq (25) holds under the common usage of $s(u, i) \in [0, 1]$, the first inequality in Eq (26) holds when $|\mathcal{I}| \geq 6$, while the second inequality in Eq (26) holds by Eq (24).

Therefore, by Eq (23) and Eq (26),

$$\mathcal{L}_{\text{AdvInfoNCE}} \geq DCG(\pi_s(u, \mathcal{I}), \mathbf{y}) \geq \exp(-\mathcal{L}_{\text{AdvInfoNCE}}). \tag{27}$$

Consequently, minimizing $\mathcal{L}_{\text{AdvInfoNCE}}$ is equivalent to minimizing $DCG(\pi_s(u, \mathcal{I}), \mathbf{y})$.

## C   Experiments

### C.1   Experimental Settings

**Datasets**

- **KuaiRec** [56] is a real-world dataset sourced from the recommendation logs of KuaiShou, a platform for sharing short videos. The unbiased testing data consist of dense ratings from 1411 users for 3327 items, with the training data being relatively sparse. We categorize items that have a viewing duration exceeding twice the length of the short video as positive interactions.

- **Yahoo!R3** [57] is a dataset that encompasses ratings for songs. The training set is comprised of 311,704 user-selected ratings ranging from 1 to 5. The test set includes ratings for ten songs randomly exposed to each user. Interactions with items receiving a rating of 4 or higher are considered positive in our experiments.

- **Coat** [58] records online shopping interactions of customers purchasing coats. The training set, characterized as a biased dataset, comprises ratings provided by users for 24 items they have chosen. The test set, on the other hand, contains ratings for 16 coats that were randomly exposed to each user. Ratings in Coat follow a 5-point scale, and interactions involving items with a rating of 4 or above are classified as positive instances in our experiments.

- **Tencent** [10] is collected from the Tencent's short-video platform. We sort the items according to their popularity in descending order and divide them into 50 groups. Each group, defined by its popularity rank, is assigned a certain number of interactions, denoted by $N_i$, for inclusion in the test set. The quantity $N_i$ is calculated based on $N_0 \cdot \gamma^{-\frac{i-1}{49}}$, where $N_0$ is the maximum number of interactions across all test groups and $\gamma$ denotes the extent of the long-tail distribution. A lower value of gamma indicates a stronger deviation from the original distribution, thus yielding a more evenly distributed test set. To ensure that the validation set mirrored the long-tail distribution of

Table 3: Dataset statistics.

|  | KuaiRec | Yahoo!R3 | Coat | Tencent |
|---|---|---|---|---|
| #Users | 7,176 | 14,382 | 290 | 95,709 |
| #Items | 10,728 | 1,000 | 295 | 41,602 |
| #Interactions | 1,304,453 | 129,748 | 2,776 | 2,937,228 |
| Density | 0.0169 | 0.0090 | 0.0324 | 0.0007 |

Table 4: The performance comparison on unbiased datasets over the MF backbone. The improvement achieved by AdvInfoNCE is significant ($p$-value $<< 0.05$).

| | Yahoo!R3 | | Coat | |
|---|---|---|---|---|
| | Recall | NDCG | Recall | NDCG |
| BPR (Rendle et al., 2012) | 0.1189 | 0.0546 | 0.2782 | 0.1748 |
| InfoNCE (van den Oord et al., 2018) | 0.1478 | 0.0694 | 0.2683 | 0.1961 |
| CCL (Mao et al., 2021) | $0.1458^{-1.35\%}$ | $0.0689^{-0.72\%}$ | $0.2682^{-0.04\%}$ | $0.1712^{-12.70\%}$ |
| BC Loss (Zhang et al., 2022) | $0.1492^{+0.95\%}$ | $\underline{0.0698}^{+0.58\%}$ | $0.2698^{+0.56\%}$ | $0.1959^{-0.10\%}$ |
| Adap-$\tau$ (Chen et al., 2023) | $\underline{0.1512}^{+2.30\%}$ | $0.0694^{+0.43\%}$ | $\underline{0.2712}^{+1.08\%}$ | $\underline{0.1986}^{+1.27\%}$ |
| **AdvInfoNCE** | $\mathbf{0.1523}*^{+3.04\%}$ | $\mathbf{0.0710}*^{+2.31\%}$ | $\mathbf{0.2905}*^{+8.27\%}$ | $\mathbf{0.1999}*^{+1.94\%}$ |

the training set and no side information of the test set is leaked, the remaining interactions are randomly divided into training and validation sets at a ratio of 60:10.

**Baselines.** In this study, we conduct a comprehensive evaluation of AdvInfoNCE using two widely adopted collaborative filtering backbones, MF [50] and LightGCN [51]. We thoroughly compare AdvInfoNCE with two categories of the latest CL-based CF methods: augmentation-based baselines (SGL [13], NCL [52], XSimGCL [27]) and loss-based baselines (CCL [53], BC Loss [54], Adap-$\tau$ [55]).

- **SGL** [13] leverages data-heuristic graph augmentation techniques to generate augmented views. It then employs contrastive learning on the augmented views and the original embeddings.

- **NCL** [52] implements contrastive learning on two types of neighbors: structural neighbors and semantic neighbors. Structural neighbors are represented by the embeddings derived from even-numbered layers in a Graph Neural Network (GNN). Semantic neighbors comprise nodes with similar features or preferences, clustered through the Expectation-Maximization (EM) algorithm.

- **XSimGCL** [27] directly infuses noise into graph embeddings from the mid-layer of LightGCN to generate augmented views. This method arises from experimental observations indicating that CF models are relatively insensitive to graph augmentation. Instead, the key determinant of their performance lies in the application of contrastive loss.

- **CCL** [53] proposes a variant of contrastive loss based on cosine similarity. Specifically, it implements a strategy for filtering out negative samples lacking substantial information by employing a margin, denoted as $m$.

- **BC Loss** [54] integrates a bias-aware margin into the contrastive loss to alleviate popularity bias. Specifically, the bias-aware margin is learned via a specialized popularity branch, which only utilizes the statistical popularity of users and items to train an additional CF model.

- **Adap-$\tau$** [55] proposes to automatically search the temperature of InfoNCE. Moreover, a fine-grained temperature is assigned for each user according to their previous loss.

**Evaluation Metrics.** We apply the all-ranking strategy, in which all items, with the exception of the positive ones present in the training set, are ranked by the collaborative filtering model for each user. An exception is KuaiRec, where the unbiased test set clusters in a small matrix [56]. This unique structure leads to a failure of the all-ranking strategy as the test set is not randomly selected from the whole user-item matrix. Consequently, for KuaiRec, we rank only the 3327 fully exposed items during the testing phase.

## C.2 Performance over the MF Backbone

Given that the implementation of augmentation-based methods is tied to the LightGCN architecture, we compare AdvInfoNCE using the Matrix Factorization (MF) backbone against only loss-based methods. As shown in Table 4 and 5, AdvInfoNCE consistently surpasses all collaborative filtering baselines with modified contrastive losses. Moreover, similar to the trend observed with the LightGCN backbone, AdvInfoNCE excels on test sets exhibiting higher distribution shifts, while still preserving remarkable performance on the in-distribution validation set. The superior performance of AdvInfoNCE across both the LightGCN and MF backbones emphasizes its model-agnostic characteristic. We advocate for considering AdvInfoNCE as a standard loss in recommender systems.

Table 5: The performance comparison on the Tencent dataset over the MF backbone. The improvement achieved by AdvInfoNCE is significant ($p$-value $<< 0.05$).

| | $\gamma = 200$ | | | $\gamma = 10$ | | | $\gamma = 2$ | | | Validation |
|---|---|---|---|---|---|---|---|---|---|---|
| | HR | Recall | NDCG | HR | Recall | NDCG | HR | Recall | NDCG | NDCG |
| BPR (Rendle et al., 2012) | 0.0835 | 0.0299 | 0.0164 | 0.0516 | 0.0190 | 0.0102 | 0.0357 | 0.0141 | 0.008 | 0.0533 |
| InfoNCE (van den Oord et al., 2018) | 0.1476 | 0.0538 | 0.0318 | 0.0920 | 0.0334 | 0.0194 | 0.0627 | 0.0233 | 0.0141 | 0.0856 |
| CCL (Mao et al., 2021) | 0.1395 | 0.0523 | 0.0317 | 0.0930 | 0.0353 | 0.0221 | 0.0683 | 0.0266 | 0.0170 | 0.0782 |
| BC Loss (Zhang et al., 2022) | 0.1546 | 0.0575 | 0.0349 | 0.1011 | 0.0378 | 0.0228 | 0.0737 | 0.0280 | 0.0178 | 0.0864 |
| Adap-$\tau$ (Chen et al., 2023) | 0.1398 | 0.0512 | 0.0302 | 0.0876 | 0.0316 | 0.0182 | 0.0591 | 0.0221 | 0.0134 | 0.0844 |
| **AdvInfoNCE** | **0.1606*** | **0.0595*** | **0.0355*** | **0.1111*** | **0.0412*** | **0.0249*** | **0.0813*** | **0.0308*** | **0.0189*** | 0.0860 |
| Imp.% over the strongest baseline | 3.87% | 3.52% | 1.86% | 9.86% | 8.86% | 9.38% | 10.33% | 9.87% | 6.28% | – |
| Imp.% over InfoNCE | 8.84% | 10.51% | 11.77% | 20.81% | 23.52% | 28.41% | 29.64% | 31.97% | 33.69% | – |

Table 6: The performance comparison on the Tencent dataset over extensive backbones. The improvement achieved by AdvInfoNCE is significant ($p$-value $<< 0.05$).

| | $\gamma = 200$ | | | $\gamma = 10$ | | | $\gamma = 2$ | | | Validation |
|---|---|---|---|---|---|---|---|---|---|---|
| | HR | Recall | NDCG | HR | Recall | NDCG | HR | Recall | NDCG | NDCG |
| UltraGCN (Mao et al., 2021) | 0.0930 | 0.0343 | 0.0203 | 0.0567 | 0.0215 | 0.0119 | 0.0400 | 0.0157 | 0.0095 | 0.0682 |
| UltraGCN + InfoNCE | 0.1436 | 0.0519 | 0.0303 | 0.0896 | 0.0324 | 0.0189 | 0.0617 | 0.0227 | 0.0135 | 0.0842 |
| UltraGCN + AdvInfoNCE | 0.1538 | 0.0569 | 0.0338 | 0.1025 | 0.0380 | 0.0227 | 0.0726 | 0.0276 | 0.0168 | 0.0883 |
| VGAE (Kipf and Welling, 2016) + InfoNCE | 0.1482 | 0.0536 | 0.0315 | 0.0923 | 0.0338 | 0.0202 | 0.0640 | 0.0237 | 0.0141 | 0.0823 |
| VGAE + AdvInfoNCE | **0.1588*** | **0.0589*** | **0.0353*** | **0.1069*** | **0.0395*** | **0.0239*** | **0.0778*** | **0.0296*** | **0.0182*** | 0.0871 |

## C.3 Performance over Extensive Backbones

To validate the generalization ability of AdvInfoNCE, we conducted experiments on additional backbones, including UltraGCN [69] and an adapted version of VGAE [70]. The results in Table 6 indicate that AdvInfoNCE performs excellently across various backbones, which showcases the generalization ability of AdvInfoNCE.

## C.4 Performance Comparison with Extensive Baselines

We compare AdvInfoNCE on the LightGCN backbone with extensive baselines on Tencent. The results in Table 7 show that AdvInfoNCE also outperforms almost all the latest debiasing [10, 71] and hard negative mining algorithms [72].

## C.5 Training Cost

Let n be the number of items, d be the embedding size, N be the number of negative sampling, M = $|\mathcal{O}^+|$ be the number of observed interactions, B be the batch size and $N_b$ be the number of mini-batches within one batch. In AdvInfoNCE, the similarity calculation for one positive item with N negative items costs $O((N + 1)d)$, and the hardness calculation costs $O(Nd)$. The total training costs of one epoch without backward propagation are summarized in Table 8. The training cost of AdvInfoNCE is a little higher than BPR loss, sharing the same complexity with InfoNCE.

In Table 9, we present both the per-epoch and total time costs for each baseline model on the Tencent dataset. As evidenced, augmentation-based contrastive learning (CL) baselines significantly cut down the overall training time, while loss-based CL baselines exhibit a complexity similar to that of InfoNCE. Surprisingly, compared to InfoNCE, AdvInfoNCE introduces only a marginal increase in computational complexity during the training phase.

# D Discussion about AdvInfoNCE

## D.1 Algorithm

Algorithm 1 depicts the detailed procedure of AdvInfoNCE. Here we uniformly sample $N$ negative items for each observed interaction and multiply a large weighting parameter $K$ in front of each negative item, as a surrogate of the whole negative set $\mathcal{N}_u$. Specifically, we adversarially train the hardness $\delta_j^{(u,i)}$ at a fixed interval before reaching the maximum adversarial training epochs $E_{adv}$. The precise methods for computing the hardness $\delta_j^{(u,i)}$ are further discussed in Section D.4.

Table 7: The performance comparison on the Tencent dataset with extensive baselines. The improvement achieved by AdvInfoNCE is significant ($p$-value $\ll 0.05$).

| | $\gamma = 200$ | | | $\gamma = 10$ | | | $\gamma = 2$ | | | Validation |
| | HR | Recall | NDCG | HR | Recall | NDCG | HR | Recall | NDCG | NDCG |
|---|---|---|---|---|---|---|---|---|---|---|
| XIR (Chen et al., 2022) | 0.1463 | 0.0538 | 0.0326 | 0.0936 | 0.0341 | 0.0211 | 0.0642 | 0.0245 | 0.0154 | 0.0883 |
| sDRO (Wen et al., 2022) | 0.1455 | 0.0516 | 0.0286 | 0.0857 | 0.0304 | 0.0166 | 0.0552 | 0.0205 | 0.0110 | 0.0872 |
| InvCF (Zhang et al., 2023) | **0.1651** | **0.0605** | 0.0331 | 0.1061 | 0.0386 | 0.0204 | 0.0722 | 0.0272 | 0.0149 | 0.0912 |
| AdvInfoNCE | 0.1600 | 0.0594 | **0.0356*** | **0.1087*** | **0.0403*** | **0.0243*** | **0.0774*** | **0.0295*** | **0.0180*** | 0.0879 |

Table 8: Time Complexity

| | +BPR | +InfoNCE | +CCL | +BC Loss | +Adap $\tau$ | +AdvInfoNCE |
|---|---|---|---|---|---|---|
| Backbone | $O(N_bBd)$ | $O(N_bB(N+1)d)$ | $O(N_bB(N+1)d)$ | $O(N_bB(N+1)d)$ | $O(N_bB(N+1)d+(M+n)d)$ | $O(N_bB(N+1)d)$ |

## D.2 Effect of the Fine-grained Hardness on KuaiRec

We conduct the same experiments as Section 4.2.1 on KuaiRec, to investigate how the fine-grained hardness affects the out-of-distribution performance.

- We plot the average value of $p(j|(u, i))$ across one batch, as depicted in Figure 4a. The figure reveals that AdvInfoNCE learns a skewed negative sampling distribution, mirroring the trend observed in the Tencent dataset. Such a distribution places more emphasis on popular negative items and reduces the difficulty of unpopular negative items, which have a higher probability of being false negatives.

- To examine the effect of fine-grained hardness, we conduct experiments with four different hardness learning strategies, including AdvInfoNCE, InfoNCE, InfoNCE-Rand, and AdvInfoNCE-Reverse. AdvInfoNCE-Reverse refers to a strategy where hardness is learned by minimizing, rather than maximizing, the loss function. This inversion results in what we term 'reversed hardness', in contrast to the approach of our AdvInfoNCE method. InfoNCE-Rand denotes the assignment of uniformly random hardness for each negative item. We conduct 5-fold experiments with different random seeds for each strategy and report the mean value with standard error in Figure 4c. As the result shows, AdvInfoNCE yields consistent improvements over the other three different hardness strategies during the training phase. In contrast, the performance of AdvInfoNCE-Reverse drops rapidly as continuously training the reversed hardness. The sustained superior performance of AdvInfoNCE indicates that it effectively promotes the generalization ability of the CF model by automatically distinguishing false negatives and hard negatives.

- Figure 4b illustrates the uniform and align loss during the training phase following the initial warm-up epochs. As demonstrated, after the warm-up phase, both InfoNCE and InfoNCE-Rand exhibit a slight increase in align loss, while their uniform loss maintains a stable level. In contrast, AdvInfoNCE significantly improves uniformity at an acceptable cost of increasing align loss. On the other hand, employing reversed hardness (as in AdvInfoNCE-Reverse) appears to have a negative impact on representation uniformity. These findings underscore the importance of fine-grained hardness in AdvInfoNCE, suggesting that AdvInfoNCE learns more generalized representations.

## D.3 Effect of the Adversarial Training Epochs on KuaiRec

In this section, we conduct experiments on KuaiRec, where AdvInfoNCE is trained for varying numbers of adversarial epochs. We plot performance metrics (Recall@20 in Figure 5a and NDCG@20 in Figure 5b) on the test set, which represents out-of-distribution data, throughout the training phase. The green stars mark the corresponding endpoint of adversarial training. As illustrated in Figure 5a and 5b, both Recall@20 and NDCG@20 show a proportional trend with the number of adversarial training epochs, up to a certain threshold. However, it is worth noting that in the extreme condition when the number of adversarial training epochs exceeds the threshold, performance on out-of-distribution sharply declines. This indicates a need to strike a balance when determining the appropriate number of adversarial training epochs for AdvInfoNCE.

Table 9: Training cost on Tencent (seconds per epoch/in total).

|  | +InfoNCE | +SGL | +NCL | +XSimGCL | +CCL | +BC loss | +Adap-$\tau$ | +AdvInfoNCE |
|---|---|---|---|---|---|---|---|---|
| MF | 16.8 / 7,123 | – | – | – | 17.2 / 4,111 | 19.1 / 6,751 | 22.3 / 7,694 | 21.6 / 11,534 |
| LightGCN | 41.2 / 21,177 | 82.5 / 4,868 | 54.9 / 5,161 | 42.1 / 842 | 42.1 / 11,114 | 43.5 / 23,664 | 55.6 / 17,236 | 44.6 / 21,586 |

---

**Algorithm 1** AdvInfoNCE

---

**Input:** observed interactions $\mathcal{O}^+$, unobserved interactions $\mathcal{O}^-$, learning rate of adversarial training $lr_{adv}$, maximum adversarial training epochs $E_{adv}$, adversarial training intervals $T_{adv}$, parameters of the CF model $\theta$, parameters of the hardness evaluation models $\theta_{adv}$, weighting parameter $K$
**Output:** $\theta$
**Initialize:** Initialize $\theta$ and $\theta_{adv}$, $e \leftarrow 1$, $e_{adv} \leftarrow 1$
**repeat**
    Freeze parameters of the hardness evaluation model $\theta_{adv}$
    Randomly sample $N$ negative items from $\mathcal{I}_u^-$ for each interaction within a batch
    Compute $s(u,i)$, $\delta_j^{(u,i)}$ with $\theta$ and $\theta_{adv}$, respectively
    Compute $\mathcal{L}_{\text{Adv}}(u,i) = -\log \frac{\exp\left(s(u,i)\right)}{\exp\left(s(u,i)\right) + K \sum_{j=1}^{N} \exp(\delta_j^{(u,i)}) \exp(s(u,j))}$
    Update $\theta$ by minimizing $\mathcal{L}_{\text{Adv}}(u,i)$
    **if** $e \bmod T_{adv} == 0$    &    $e_{adv} \leq E_{adv}$ **then**
        Freeze parameters of the CF model $\theta$
        Update $\theta_{adv}$ by maximizing $\mathcal{L}_{\text{Adv}}(u,i)$
        $e_{adv} \leftarrow e_{adv} + 1$
    **end if**
    $e \leftarrow e + 1$
**until** CF model converges

---

### D.4 Hardness Learning Strategy

To accurately evaluate the hardness of each negative instance, we need to establish a mapping from unobserved user-item pairs to their corresponding hardness values, and this mapping mechanism can be diversified. Generally, the hardness learning strategy can be formulated as:

$$\delta_j^{(u,i)} \doteq \log(|\mathcal{N}_u| \cdot p(j|(u,i))) \tag{28}$$

$$\doteq \log\left(|\mathcal{N}_u| \cdot \frac{\exp\left(g_{\theta_{adv}}(u,j)\right)}{\sum_{k=1}^{|\mathcal{N}_u|} \exp(g_{\theta_{adv}}(u,k))}\right), \tag{29}$$

where $g_{\theta_{adv}}(u,j)$ is a raw hardness score function for the unobserved user-item pair $(u,j)$, and $p(j|(u,i))$ is the probability of sampling the negative instance $j$, which is calculated by normalizing $g_{\theta_{adv}}(u,j)$. In this paper, we proposed two specific mapping methods: embedding-based (*i.e.,* AdvInfoNCE-embed) mapping and multilayer perceptron-based (*i.e.,* AdvInfoNCE-mlp) mapping. It should be noted that all the results reported in the main text of this paper are based on the implementation of the AdvInfoNCE-embed version.

**AdvInfoNCE-embed.** The hardness computation process in AdvInfoNCE-embed follows a similar protocol as CF models. We directly map the index of users and items into its corresponding hardness embedding and calculate the hardness of each user-item pair through a score function. Specifically, this process involves a user hardness encoder $\psi_{\theta_{adv}}(\cdot) : \mathbb{U} \to \mathbb{R}^d$ and an item hardness encoder $\phi_{\theta_{adv}}(\cdot) : \mathbb{I} \to \mathbb{R}^d$, where $\theta_{adv}$ denotes all the trainable parameters of the hardness learning model. In our experiments, we adopt the same embedding dimension as the CF models for hardness learning. For the score function, we define $g_{\theta_{adv}}(u,j) = \psi_{\theta_{adv}}(u)^\top \cdot \phi_{\theta_{adv}}(j)$.

**AdvInfoNCE-mlp.** Unlike AdvInfoNCE-embed, AdvInfoNCE-mlp maps the embeddings of items and users from the CF model into another latent space with two multilayer perceptrons (MLPs) and calculates the hardness on this latent space. By respectively defining the MLPs for users and items as $\text{MLP}_\text{u}$ and $\text{MLP}_\text{v}$, the score function for hardness calculation is defined as $g_{\theta_{adv}}(u,j) = \text{MLP}_\text{u}\left(\psi_\theta(u)\right)^\top \cdot \text{MLP}_\text{v}\left(\phi_\theta(j)\right)$. In our experiment, we simply employ one-layer MLPs and set the dimension of latent space as four. It's worth noting that this MLP-based implementation of

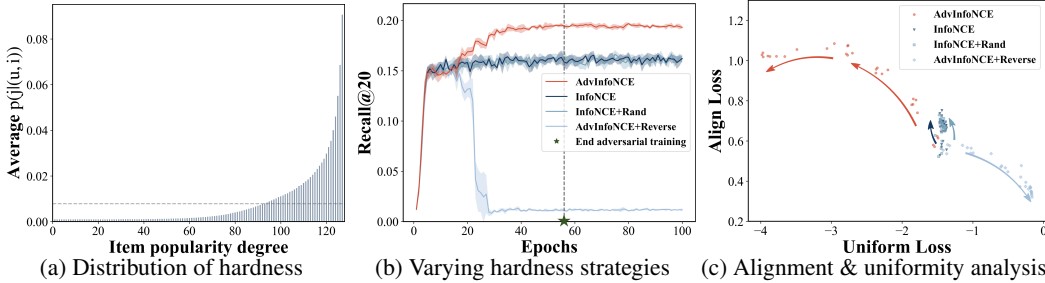

| (a) Distribution of hardness | (b) Varying hardness strategies | (c) Alignment & uniformity analysis |

Figure 4: Study of hardness. (4a) Illustration of hardness *i.e.,* the probability of negative sampling $(p\,(j\,|\,(u,i)))$ learned by AdvInfoNCE *w.r.t.* item popularity on KuaiRec. The dashed line represents the uniform distribution. (4b) Performance comparisons with varying hardness learning strategies on KuaiRec. (4c) The trajectories of align loss and uniform loss during training progress. Lower values indicate better performance. Arrows denote the losses' changing directions.

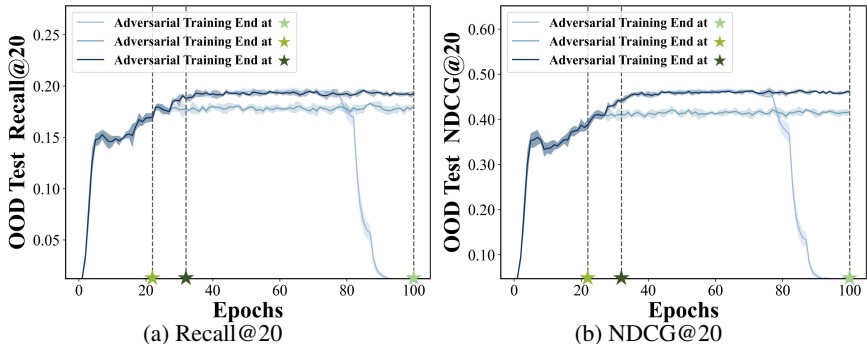

(a) Recall@20       (b) NDCG@20

Figure 5: The performance on KuaiRec with different numbers of adversarial training epochs. (5a) Performance comparisons *w.r.t.* Recall@20 on KuaiRec. (5b) Performance comparisons *w.r.t.* NDCG@20 on KuaiRec.

AdvInfoNCE may also be adaptable for handling out-of-distribution tasks in other fields, such as computer vision (CV) and natural language processing (NLP).

As reported in Table 10, both AdvInfoNCE-embed and AdvInfoNCE-mlp yield significant improvements over InfoNCE. Moreover, AdvInfoNCE-embed generally outperforms AdvInfoNCE-mlp.

### D.5 The Intuitive Understanding of AdvInfoNCE

In this section, we aim to understand the mechanism of AdvInfoNCE intuitively, from the perspective of false negative identification.
Figure 6 illustrates the changes of the out-of-distribution performance and FN identification rate during the training process on Tencent. Here, the FN identification rate indicates the proportion of false negatives with negative $\delta_j$. It can be observed that the out-of-distribution performance exhibits a rising trend along with the FN identification rate. Meanwhile, the out-of-distribution performance of InfoNCE remains relatively low. This indicates that AdvInfoNCE enhances the generalization ability of the CF model by identifying false negatives.

Figure 7 illustrates how AdvInfoNCE adjusts the scores and rankings of sampled negative items, by identifying the false negatives and true negatives. We retrieve the negative items sampled during training. If a sampled negative item appears in the test set, it is labeled as a false negative (FN); otherwise. In Figure 7a and 7b, the leftmost item represents a false negative, while the other two items on the right are negatives. The bar charts in blue and red depict the cosine similarity scores of sampled negative items measured by InfoNCE and AdvInfoNCE respectively. The rankings of sampled negative items are annotated above the bars. The line graph illustrates the hardness $\delta_j$ computed by AdvInfoNCE, measured on the right axis. It can be observed that when the hardness $\delta_j$

Table 10: The performance comparison between AdvInfoNCE-embed and AdvInfoNCE-mlp over the LightGCN backbone.

| | KuaiRec | | Yahoo!R3 | | Coat | | Tencent ($\gamma = 200$) | | Tencent ($\gamma = 10$) | | Tencent ($\gamma = 2$) | |
| | Recall | NDCG | Recall | NDCG | Recall | NDCG | Recall | NDCG | Recall | NDCG | Recall | NDCG |
|---|---|---|---|---|---|---|---|---|---|---|---|---|
| InfoNCE | 0.1800 | 0.4529 | 0.1475 | 0.0698 | 0.2689 | 0.1882 | 0.0540 | 0.0320 | 0.0332 | 0.0195 | 0.0242 | 0.0145 |
| **AdvInfoNCE-embed** | 0.1979 | 0.4697 | 0.1527 | 0.0718 | 0.2846 | 0.2026 | 0.0594 | 0.0356 | 0.0403 | 0.0243 | 0.0295 | 0.0180 |
| Imp.% over InfoNCE | 9.94% | 3.71% | 3.53% | 2.87% | 5.84% | 7.65% | 10.00% | 11.25% | 21.39% | 24.62% | 21.90% | 24.14% |
| **AdvInfoNCE-mlp** | 0.1851 | 0.4579 | 0.1545 | 0.0724 | 0.2843 | 0.2002 | 0.0567 | 0.0339 | 0.0364 | 0.0221 | 0.0260 | 0.0160 |
| Imp.% over InfoNCE | 2.83% | 1.10% | 4.75% | 3.72% | 5.73% | 6.38% | 5.00% | 5.94% | 9.64% | 13.33% | 7.44% | 10.34% |

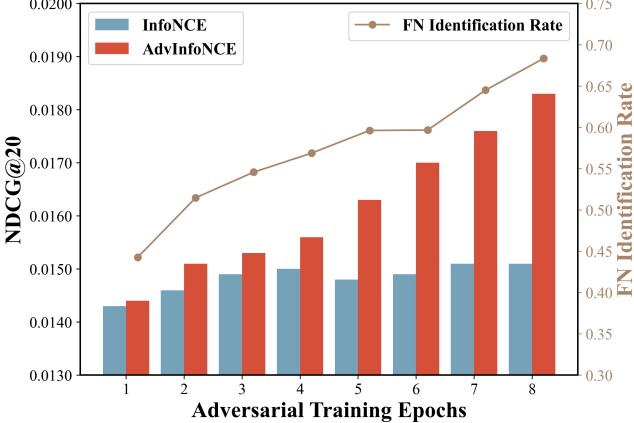

Figure 6: FN identification rate and NDCG@20 during training on Tencent, where FN identification rate indicates the proportion of false negatives (FN) with negative $\delta_j$ and NDCG@20 shows the out-of-distribution performance. As training proceeds, AdvInfoNCE' FN identification rate increases, capping at nearly 70%. This reveals AdvInfoNCE's capability to identify approximately 70% of false negatives in the test set. We attribute the superior recommendation performance of AdvInfoNCE over InfoNCE to this gradual identification.

is negative (*i.e.,* indicating that the item is identified as a false negative), the cosine similarity score improves. Conversely, if the hardness $\delta_j$ is positive, the score decreases.

## E  Hyperparameter Settings

For a fair comparison, we conduct all the experiments in PyTorch with a single Tesla V100-SXM3-32GB GPU and an Intel(R) Xeon(R) Gold 6248R CPU. We optimize all methods with the Adam optimizer and set the layer numbers of LigntGCN by default at 2, with the embedding size as 64 and the weighting parameter $K$ as 64. We search for hyperparameters within the range provided by the corresponding references. For AdvInfoNCE, we search $lr_{adv}$ in [1e-1, 1e-4], $E_{adv}$ in [1, 30] and $T_{adv}$ in {5, 10, 15, 20}. We adopt the early stop strategy that stops training if Recall@20 on the validation set does not increase for 20 successive evaluations. It's worth noting that AdvInfoNCE inherits the hyperparameter sensitivity property of adversarial learning, therefore it's necessary to choose proper hyperparameters for different datasets. We suggest selecting a suitable adversarial learning rate $lr_{adv}$ first and then increasing the number of adversarial training epochs $E_{adv}$ gradually until AdvInfoNCE reaches a relatively stable performance. We report the effect of changing the number of negative sampling in Table 13, where $N$ is the number of negative sampling.

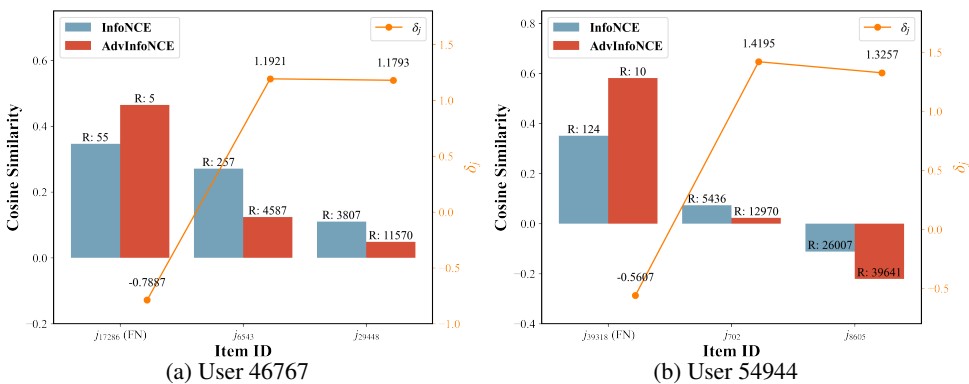

(a) User 46767          (b) User 54944

Figure 7: Case studies of refining the item ranking. With two randomly sampled users along with their sampled negative items, we subsequently retrieve their associated $\delta$ values, ranking positions, and cosine similarities. Here FN denotes false negative (*i.e.,* interactions unobserved during training but present in testing). The bar charts demonstrate the cosine similarity scores of these sampled negative items as gauged by both InfoNCE and AdvInfoNCE. Their rankings are annotated atop the bars. An accompanying line illustrates the hardness $\delta_j$ derived by AdvInfoNCE (measured on the right axis). Notably, when $\delta < 0$, AdvInfoNCE identifies and elevates an FN; conversely, for a potentially true negative, AdvInfoNCE leans towards a positive $\delta$ and declines its rank.

Table 11: Hyperparameters search spaces for baselines.

| | Hyperparameter space |
|---|---|
| **MF** & **LightGCN** | lr $\sim$ {1e-5, 3e-5, 5e-5, 1e-4, 3e-4, 5e-4, 1e-3}, batch size $\sim$ {64, 128, 256, 512, 1024, 2048} No. negative samples $\sim$ {64, 128, 256, 512} |
| **SSM** | $\tau \sim [0.05, 3]$ |
| **CCL** | $w \sim$ {1, 2, 5, 10, 50, 100, 200}, $m \sim$ {0.2, 0.4, 0.6, 0.8, 1} |
| **BC Loss** | $\tau_1 \sim [0.05, 3]$, $\tau_2 \sim [0.05, 3]$ |
| **Adap-$\tau$** | $warm\_up\_epochs \sim$ {10, 20, 50, 100} |
| **SGL** | $\tau \sim [0.05, 3]$, $\lambda_1 \sim$ {0.005, 0.01, 0.05, 0.1, 0.5, 1.0}, $\rho \sim$ {0, 0.1, 0.2, 0.3, 0.4, 0.5} |
| **NCL** | $\tau \sim [0.05, 3]$, $\lambda_1 \sim$ [1e-10, 1e-6], $\lambda_2 \sim$ [1e-10, 1e-6], $k \sim$ [5, 10000] |
| **XSimGCL** | $\tau \sim [0.05, 3]$, $\epsilon \sim$ {0.01, 0.05, 0.1, 0.2, 0.5, 1.0}, $\lambda \sim$ {0.005, 0.01, 0.05, 0.1, 0.5, 1.0}, $l* = 1$ |

Table 12: Model architectures and hyperparameters for AdvInfoNCE.

| | $lr_{adv}$ | $E_{adv}$ | $T_{adv}$ | $\tau$ | lr | batch size | No. negative samples |
|---|---|---|---|---|---|---|---|
| **LightGCN** | | | | | | | |
| Tencent | 5e-5 | 7 | 5 | 0.09 | 1e-3 | 2048 | 128 |
| KuaiRec | 5e-5 | 12 | 5 | 2 | 3e-5 | 2048 | 128 |
| Yahoo!R3 | 1e-4 | 13 | 5 | 0.28 | 5e-4 | 1024 | 64 |
| Coat | 1e-2 | 20 | 15 | 0.75 | 1e-3 | 1024 | 64 |
| **MF** | | | | | | | |
| Tencent | 5e-5 | 8 | 5 | 0.09 | 1e-3 | 2048 | 128 |
| Yahoo!R3 | 1e-4 | 12 | 5 | 0.28 | 5e-4 | 1024 | 64 |
| Coat | 1e-2 | 18 | 15 | 0.75 | 1e-3 | 1024 | 64 |

Table 13: Varying number of negative sampling on Tencent

| | $\gamma = 200$ | | | $\gamma = 10$ | | | $\gamma = 2$ | | | Validation |
|---|---|---|---|---|---|---|---|---|---|---|
| N | HR | Recall | NDCG | HR | Recall | NDCG | HR | Recall | NDCG | NDCG |
| 64 | 0.1513 | 0.0563 | 0.0333 | 0.1006 | 0.0373 | 0.0225 | 0.0708 | 0.0269 | 0.0164 | 0.0854 |
| 128 | 0.1600 | 0.0594 | 0.0356 | 0.1087 | 0.0403 | 0.0243 | 0.0774 | 0.0295 | 0.0180 | 0.0879 |
| 256 | 0.1642 | 0.0609 | 0.0367 | 0.1125 | 0.0419 | 0.0253 | 0.0815 | 0.0310 | 0.0189 | 0.0889 |

