# OpenReview forum: "Empowering Collaborative Filtering with Principled Adversarial Contrastive Loss"
_NeurIPS.cc/2023/Conference — NeurIPS 2023 poster_

### Official Review · Reviewer_1hVj · 2023-06-29

**Soundness:** 3 good
**Presentation:** 3 good
**Contribution:** 3 good
**Rating:** 5
**Confidence:** 3

**Summary:**

In this work, the authors focus on improving the generalization ability of the top-K recommendation model by a proposed principled Adversarial InfoNCE loss (AdvInfoNCE). Existing contrastive learning based methods usually lack considering the tailored inductive bias (such as hard negatives and false negatives) and sufficient theoretical understanding for the generalization ability. The proposed AdvInfoNCE loss could adaptively explore and assign hardness (weight) for each negative instance in an adversarial fashion. The theoretical guarantees and experiments demonstrate the effectiveness of the proposed AdvInfoNCE loss.

**Strengths:**

1.	The motivation that adaptively assigns hardness for each negative instance in an adversarial fashion to improve the generalization ability is justified.
2.	The theoretical proof demonstrates the generalization ability of the proposed AdvInfoNCE loss.
3.	The experiments on unbiased datasets and out-of-distribution settings demonstrate the effectiveness of the proposed AdvInfoNCE loss in terms of generalization ability.


**Weaknesses:**

1.	It would be better to add an intuitive example to show the adaptive learning process for hard negatives and false negatives in the min and max stages.
2.	In terms of adaptively learning hardness for each negative instance, it would be better to add some baselines focusing on mining hard negatives or false negatives.
3.	In terms of the generalization ability, it would be better to add some baselines focusing on out-of-distribution. Besides, OOD experiments are also performed on popularity-based distribution shift scenarios, what’s the difference compared with debiasing experiments?


**Questions:**

1.	Could you please give an intuitive description to show the adaptive learning process for hard negatives and false negatives in the min and max stage ? Especially for hard negatives and false negatives?
2.	Why the corresponding item is identified as a hard negative rather than a general negative when sigma > 0 ?
3.	In terms of adaptively learning hardness for each negative instance, could you please add some baselines focusing on mining hard negatives or false negatives [1,2] ?
4.	In terms of the generalization ability, could you please add some baselines focusing on out-of-distribution [3,4] ?

[1] Chen, Jin, et al. "Cache-Augmented Inbatch Importance Resampling for Training Recommender Retriever." In NIPS, 2022.
[2] Chen, Jin, et al. "Learning Recommenders for Implicit Feedback with Importance Resampling." In WWW, 2022.
[3] Hongyi Wen, et al. "Distributionally-robust Recommendations for Improving Worst-case User Experience". In WWW, 2022.
[4] An Zhang, et al. "Invariant Collaborative Filtering to Popularity Distribution Shift". In WWW, 2023.


**Limitations:**

The authors adequately point out the limitations and there is no negative societal impact of their work.

---

> ### Author Rebuttal · Authors · 2023-08-10
>
> **Response to Reviewer $\color{green}{\text{1hVj}}$**
>
> We gratefully thank you for the valuable comments. To address your concerns, below we provide the point-to-point responses.
>
> >**Comment 1 + Question 1: Intuitive example**
>
> We value your insightful comments. To better clarify the role of $\delta$ and its learning process for hard and false negatives, we **conducted additional illustrative examples**.
>
> The illustrative example should highlight two points. 1. AdvInfoNCE could effectively **identify the false and hard negatives** via learnable $\delta_j$ (max-stage). 2. $\delta$ helps to **refine the item ranking** compared to InfoNCE (min-stage).
> - **Identification of False and Hard Negatives:**
> On the Tencent training data, we trained both the InfoNCE and AdvInfoNCE models. Interactions unobserved during training but present in testing are labeled as false negatives (FN), otherwise true negatives (TN).
> Based on our theoretical assumption, for a FN, we wish a more relaxed constraint, leading to $\delta < 0$.
> To validate this assumption, we introduce the 'FN identification rate', a metric determining the proportion of FNs where $\delta < 0$.
> As Fig 6 in one-page uploaded pdf shows, our observations are consistent with our claim.
> As training proceeds, the FN identification rate increases, capping at nearly 70%.
> This reveals AdvInfoNCE's capability to identify approximately 70% of FNs in test set.
> We attribute the superior performance of AdvInfoNCE over InfoNCE to this gradual identification.
> - **Refinement of Item Ranking:**
> we randomly draw two users along with their FN and TN items, subsequently retrieving their associated $\delta$ values, ranking positions, and cosine similarities, as demonstrated in Fig 7.
> Consistent with our prior findings, for an FN, AdvInfoNCE generally assigns a negative $\delta$.
> This negative $\delta$, indicating a more lenient feasible zone constraint, enables the recommender to achieve higher cosine similarity.
> This, in turn, escalates the FN's ranking.
> For instance, as Fig 7(a) shows, given $\delta=−0.7887$, AdvInfoNCE elevates an FN from the 55th to a commendable 5th position. Conversely, for a TN, AdvInfoNCE leans towards a positive $\delta$, narrowing the feasible zone, thus distancing it from positives.
> An exemplary case is the TN $j_{6543}$ in Fig 7(a), where AdvInfoNCE, upon learning its $\delta = 1.1921$, declines its rank from 257th to 4587th.
> Such real-world cases attest to $\delta$'s role in fine-tuning recommendation ranking.
>
> In a word, for a specific u, the learnable $\delta$ variable measures the item hardness and further frame a fine-grained ranking criterion.
>
> >**Comment 2 + Comment3: Add mining hard negatives & OOD baselines**
>
> Thanks so much for bringing these excellent works to us. Folloing your suggestion, we have **conducted additional experiments** on XIR [1], S-DRO [2], and InvCF [3] on Tencent.
> We appreciate your recommendation of including AdaSIR [4]; however, due to the constraints of the rebuttal period, we have planned to consider it as a baseline in our future work.
>
> **Table 1: Overall performance on Tencent**
> |        |        | $\gamma=200$ |        |        | $\gamma=10$ |        |        | $\gamma=2$ |        | Validation |
> | :----: | :----: | :----------: | :----: | :----: | :---------: | :----: | :----: | :--------: | :----: | :--------: |
> |        |   HR   |    Recall    |  NDCG  |   HR   |   Recall    |  NDCG  |   HR   |   Recall   |  NDCG  |    NDCG    |
> |  XIR   | 0.1463 |    0.0538    | 0.0326 | 0.0936 |   0.0341   | 0.0211 | 0.0642 |   0.0245   | 0.0154 |   0.0883   |
> |    sDRO    | 0.1455 | 0.0516 | 0.0286 | 0.0857 | 0.0304 | 0.0166 | 0.0552 | 0.0205 | 0.0110 | 0.0872 |
> |   InvCF    | **0.1651** | **0.0605** | 0.0331 | 0.1061 | 0.0386 | 0.0204 | 0.0722 | 0.0272 | 0.0149 | **0.0912** |
> | AdvInfoNCE| 0.1600 |  0.0594  | **0.0356** | **0.1087** | **0.0403**  | **0.0243** | **0.0774** | **0.0295** | **0.0180** |   0.0879   |
>
> We observe that:
> - AdvInfoNCE consistently exhibits superior performance compared to hard negative mining methods and OOD baselines across varying test set levels in terms of NDCG metric. Such results underscore the robustness and generalization ability of AdvInfoNCE.
>
> - For the case where $\gamma = 200$, although InvCF slightly edges out AdvInfoNCE in HR and Recall, a potential explanation lies in InvCF's design which selects in-batch negatives (1024 in total). In contrast, AdvInfoNCE uses only 128 negatives. As can be inferred from another analysis (Table 3 in response for the first Reviewer ikze), amplifying the number of negative sampling could boost the performance of AdvInfoNCE. (Sorry for inconvenience. Due to space constraints, I can not directly include Table here.)
>
> - Specific Strength of XIR: XIR demonstrates enhanced performance particularly when faced with data exhibiting a long-tailed distribution, i.e., $\gamma =200$. We attribute this advantage of XIR to adaptively achieve a more accurate estimation of the softmax distribution.
>
>
> >**Question 2: General or hard negative**
>
> Your question goes to the heart of our method. In our paper, we define an item with $\delta > 0$ as a hard negative based on the gradient analysis presented in Appendix B.3.
> The gradients corresponding to the negative item j have a proportional relation to the $\exp(\delta_j)$.
> In other words, for $\delta_j >0$, the recommender exhibits more attention to this item by a factor of $\exp(\delta_j) >1$.
> Such characteristics naturally fall under the category of hard negatives based on hard negative mining.
>
>
> [1] Cache-Augmented Inbatch Importance Resampling for Training Recommender Retriever. 2022
>
> [2] Distributionally-robust Recommendations for Improving Worst-case User Experience. 2022
>
> [3] Invariant Collaborative Filtering to Popularity Distribution Shift. 2023
>
> [4] Learning Recommenders for Implicit Feedback with Importance Resampling. 2022

---

> > ### Comment · Reviewer_1hVj · 2023-08-18
> >
> > Thank you for your efforts and responses. I will keep my rating.

---

> ### Author Response · Authors · 2023-08-14
> **Follow-up discussion**
>
> Thank you for your valuable feedback on our submission, particularly your suggestions to include **an illustrative example** and add **XIR, AdaSIR, S-DRO, and InvCF** as new baselines. These insightful suggestions enhance the quality of our work and better strengthen our claims. We hope that these improvements will be taken into consideration.
>
> If we fully address your concerns about our paper, we would be grateful if you could re-evaluate our paper.
> If you have additional concerns, we remain open and would be more than happy to discuss with you.

---

### Official Review · Reviewer_qmEP · 2023-07-03

**Soundness:** 4 excellent
**Presentation:** 4 excellent
**Contribution:** 3 good
**Rating:** 8
**Confidence:** 4

**Summary:**

This paper is on collaborative filtering (CF) enhanced by contrastive learning (CL). The authors point out that the adoption of CL into CF is suboptimal due to challenges such as the issue of out-of-distribution, the risk of false negatives, and the nature of top-K evaluation. They also note that current CL-based CF methods lack consideration of the tailored inductive bias for CF and have limited theoretical understanding of their generalization ability.

To address these limitations, the authors propose a principled Adversarial InfoNCE loss for CF that focuses on mining hard negatives and distinguishing false negatives from the vast unlabeled user-item interactions. The proposed method is compared with several state-of-the-art contrastive learning-based CF methods on both unbiased and synthetic datasets. The experiments show that the proposed method outperforms the baselines in terms of accuracy and robustness, demonstrating its potential for improving the performance of recommender systems.

**Strengths:**

i) The paper proposes a novel approach to adaptive contrastive learning in collaborative filtering by adopting an adversarial approach. The proposed Adversarial InfoNCE loss addresses the limitations of existing methods and allows for the fine-grained assignment of hardness to each negative user-item pair, which enhances the recommender's generalization ability and empowers top-K recommendations via informative negative sampling.

ii) The paper provides innovative theoretical insights into the benefits of adversarial hardness learning. It shows that the proposed hardness scores are correlated with the out-of-distribution problem of recommendation, and can thereby enhance the generalization ability of recommenders.

iii) The study of hardness gives an in-depth analysis on the learned hardness scores, which uncovers the importance of learning correct hardness for the massive negative samples without observations.

**Weaknesses:**

i) The proposed method can be viewed as an adaptive SSL method for recommendation [1-2]. Also, it can be a learnable negative sampling approach [3-4]. Literature review (baseline comparison would do better) should be done for these two very relevant research line.

ii) Though a brief training cost experiment is given in the appendix. I would expect more detailed efficiency experiments or analysis to support the claim that the proposed AdvInfoNCE can serve as a foundation loss for future CF researches, as the proposed approach utilizes the adversarial training method.

[1] Graph contrastive learning with adaptive augmentation

[2] Automated Self-Supervised Learning for Recommendation

[3] Personalized Ranking with Importance Sampling

[4] AHP: Learning to Negative Sample for Hyperedge Prediction

**Questions:**

It would be better for the authors to clarify the relations between the proposed method and the related research lines I mentioned above. And I would expect the authors to give a more detailed efficiency analysis or experiments.

---

> ### Author Rebuttal · Authors · 2023-08-10
>
> **Response to Reviewer $\color{purple}{\text{qmEP}}$**
>
> Thanks so much for your time and positive feedback! To address your concerns, we present the point-to-point responses as follows.
> We have carefully revised our paper, taking all your feedbacks into account.
>
>
> >**Comment 1: Clarify the relations** - "The proposed method can be viewed as an adaptive SSL method for recommendation. Also, it can be a learnable negative sampling approach. Literature review (baseline comparison would do better) should be done for these two very relevant research line."
>
> Thanks for your insightful comments. We acknowledge the importance of clarifying the relationship between our work and both the adaptive SSL method and learnable negative sampling approaches.
> Following your suggestion, we have carefully **revised the related work section** to provide a more comprehensive literature review that aligns our method with both mentioned paradigms.
> Additionally, heeding your advice, we have **incorporated two new baselines** - adaptive SSL method: AutoCF [1] and negative sampling approach: XIR [2]. The results are summarized in the table 1 provided:
>
> **Table 1: Two new baselines on Tencent**
> |        |        | $\gamma=200$ |        |        | $\gamma=10$ |        |        | $\gamma=2$ |        | Validation |
> | :----: | :----: | :----------: | :----: | :----: | :---------: | :----: | :----: | :--------: | :----: | :--------: |
> |        |   HR   |    Recall    |  NDCG  |   HR   |   Recall    |  NDCG  |   HR   |   Recall   |  NDCG  |    NDCG    |
> | AutoCF | 0.1560 |    0.0570    | 0.0317 | 0.1092 |   0.0399    | 0.0227 | 0.0797 |   0.0295   | 0.0172 |   0.0723   |
> |  XIR   | 0.1463 |    0.0538    | 0.0326 | 0.0936 |   0.0341   | 0.0211 | 0.0642 |   0.0245   | 0.0154 |   **0.0883**   |
> | AdvInfoNCE | **0.1600** |  **0.0594**  | **0.0356** | **0.1087** | **0.0403**  | **0.0243** | **0.0774** | **0.0295** |**0.0180** |   0.0879   |
>
> We observe that:
> -  Consistency in Superiority: AdvInfoNCE consistently showcases better performance over both AutoCF and XIR across various levels of out-of-distribution test sets. This demonstrates the robustness and generalization ability of our approach.
>
> - XIR's Specific Strength: XIR demonstrates enhanced performance particularly when faced with data exhibiting a long-tailed distribution. We attribute this advantage of XIR to adaptively achieve a more accurate estimation of the softmax distribution.
>
> - Impressive Performance for AutoCF: Surprisingly, in some metrics, AutoCF exceeds the performance of the second-best methods reported in our original submission. We believe the reason for AutoCF's success lies in its ability to perform masked subgraph augmentation automatically.
>
>
> These experiments have enriched our understanding of where AdvInfoNCE stands in the current research landscape. We have incorporated these insights into our revised manuscript.
>
>
>
> >**Comment 2: Efficiency analysis** - "Though a brief training cost experiment is given in the appendix. I would expect more detailed efficiency experiments or analysis to support the claim that the proposed AdvInfoNCE can serve as a foundation loss for future CF researches, as the proposed approach utilizes the adversarial training method."
>
> We highly appreciate your insightful suggestions.
> Following your suggestions, we have **analyzed the time complexity** of our proposed AdvInfoNCE method and, for a comprehensive understanding, compared it with two standard losses in CF (BPR and InfoNCE) and other Contrastive Learning-based CF methods (CCL, BC loss, Adap-$\tau$).
>
> Let's define some notations for clarity:
> - $n$: Number of users
> - $d$: Embedding size
> - $N$: Number of negative sampling
> - $M = |\mathcal{O}^{+}|$: Number of observed interactions
> - $B$: Batch size
> - $N_{b}$: Number of mini-batches within one epoch
>
> Clearly, The total time complexity of one epoch without backward propagation for BPR loss is $O(N_{b}Bd)$.
> For AdvInfoNCE, the similarity calculation for one positive item along with N negative items costs $O((N+1)d)$, and the hardness calculation costs $O(Nd)$. This implies that the training cost of AdvInfoNCE, though slightly higher than that of BPR loss, shares the same complexity order with InfoNCE.
> Detailed computational costs of  sampling based losses are summarized as follows:
>
> **Table 2: Time complexity comparison**
>
>
> | InfoNCE | CCL | BC Loss | Adap-$\tau$ | AdvInfoNCE |
> | :-----: | :-----: | :-----: | :-----: | :-----: |
> | $O(N_{b}B(N+1)d)$ | $O(N_{b}B(N+1)d)$ | $O(N_{b}B(N+1)d)$ | $O(N_{b}B(N+1)d+(M+n)d)$ | $O(N_{b}B(N+1)d)$ |
>
>
>
>
> [1] Automated Self-Supervised Learning for Recommendation. 2023
>
> [2] Cache-Augmented Inbatch Importance Resampling for Training Recommender Retriever. 2022

---

> > ### Comment · Reviewer_qmEP · 2023-08-17
> >
> > Thank you for your responses. I believe this paper is a clear accept and have raised my rating.

---

> > > ### Author Response · Authors · 2023-08-17
> > > **Thanks!**
> > >
> > > We appreciate your acknowledgment of our efforts in addressing the concerns. Your insightful comments have been instrumental in enhancing the quality of our work.

---

### Official Review · Reviewer_PLbw · 2023-07-05

**Soundness:** 3 good
**Presentation:** 3 good
**Contribution:** 4 excellent
**Rating:** 8
**Confidence:** 4

**Summary:**

This paper studies contrastive learning (CL) in collaborative filtering (CF) for top-k recommendation. In particular, it focuses on the CF-tailored challenges for CL, and then presents adversarial infoNCE (AdvInfoNCE) loss. This loss dynamically assigns hardness to negative instances and incorporates a fine-grained ranking criterion to improve the CF recommender’s generalization ability. Furthermore, this paper highlights theoretical properties of AdvInfoNCE. Experiments on both synthetic and real-world datasets are done to show the effectiveness of this loss, especially in out-of-distribution scenarios.

**Strengths:**

1.	The motivation of considering fine-grained hardness-aware ranking criteria is clear and reasonable. Technically, it is insightful and novel to transfer the distinguish between false and hard negatives into a margin learning problem.
2.	As proved in Sec 3.3, the adversarial training framework of AdvInfoNCE is natural and somehow equivalent to solve a distributionally robust optimization (DRO) problem. I appreciate this theoretical guarantee that can endow a AdvInfoNCE-trained CF models with better generalization ability.
3.	The experiments are done on four datasets with two CF base models, which are sufficient to demonstrate the effectiveness of the proposed loss. Moreover, the selected baselines are quite new, including some recent and strong works.
4.	This proposed loss seems to be a general loss that can be applied in general recommendation models.


**Weaknesses:**

1.	The hardness is denoted as $\delta$ in Sec 3.2, while related to $p(j|(u,i))$. This is not explained clearly. More clarifications are needed here.
2.	In Line 214, the limitation is stated as the ‘training instability’, which is not empirically shown in the experiments, such as indicated by ‘training loss variance’. It would be better to discuss more about this instability.
3.	Although the proof of Theorem 3.1 seems correct, the DL-divergence In Line 195 misses a minus sign. Please double check and fix it.


**Questions:**

1.	The hardness is denoted as $\delta_{j}^{(u)}$ first and then $\delta_{j}^{(u,i)}$, which seems confusing. Is it related to a specific user $u$ or a user-item pair $(u,i)$?

**Limitations:**

Please refer to the weaknesses and questions.

---

> ### Author Rebuttal · Authors · 2023-08-10
>
> **Response to Reviewer $\color{orange}{\text{PLbw}}$**
>
> We thank the reviewer for the thorough and valuable feedback. To address your concerns, we present the point-to-point responses as follows. We have carefully revised our paper by taking into account all your suggestions. Looking forward to more discussions with you.
>
> >**Comment 1: Clearer explanation** - "The hardness is denoted as $\delta$ in Sec 3.2, while related to $p(j|(u,i))$. This is not explained clearly. More clarifications are needed here."
>
> Thank you for highlighting the need for clearer explaination concerning the relation between hardness and the negative sampling probability.
> Following your suggestion, we have **revised the methodology section** in the manuscript. For the sake of addressing your question directly, we would like to provide a detailed elaboration here:
>
> In our paper, specifically at Line 195, the hardness $\delta_j^{(u,i)}$ for a negative item $j$ with respect to observed interaction $(u,i)$ is defined as:
> $\delta_j^{(u,i)} \doteq \log(|N_u|\cdot p(j|(u,i)))$.
> Here $|N_u|$ denotes the total count of negative items for user u.
> $p(j|(u,i))$ signifies the likelihood of picking item $j$ as a negative item given the observed interaction $(u,i)$.
> By defining the hardness in this manner, we closely relate our hardness to a particular strategy used for negative sampling.
>
> Moreover, as outlined in Theorem 3.1, optimizing AdvInfoNCE loss is equivalent to solving a Distributionally Robust Optimization (DRO) problem focusing on the negative sampling strategy.
> For instance, in scenarios where uniform negative sampling is employed (i.e., $p(j|(u,i)) = \frac{1}{|N_u|}$), the hardness value $\delta_j^{(u,i)}$ always equals 0.
>
> This offers an insight: InfoNCE is a special case of AdvInfoNCE, where incorporates the uniform negative sampling.
> As a result, with no difference of $\delta_j$, InfoNCE is unable to differentiate between true negatives and false negatives, which is consistent with our earlier illustration in Figure 1 (a).
>
>
> > **Comment 2: More discussion about instability** - "In Line 214, the limitation is stated as the ‘training instability’, which is not empirically shown in the experiments, such as indicated by ‘training loss variance’. It would be better to discuss more about this instability."
>
> Thanks for your valuable comments.
> The "instability" here pertains to the instability and inconsistency of hyperparameters (i.e., the learning rate and epochs for adversarial training) ranges across different datasets.
> As reported in Table 9, the optimal adversarial learning rate varies largely across different datasets (e.g., 1e-2 for Coat and 5e-5 for Tencent).
> Moreover, Figure 3 indicates that increasing the number of epochs of adversarial training without constraints leads to a significant decline in in-distribution performance.
> Additionally, when exploring the role of negative sampling numbers, we found that these two hyperparameters also require delicate adjustments for varying numbers of negative samples. Inappropriate learning rates for adversarial training can result in varying degrees of hardness mining, consequently leading to inconsistent and suboptimal performance.
>
>
> > **Comment 3: missing a minus sign** - "Although the proof of Theorem 3.1 seems correct, the DL-divergence In Line 195 misses a minus sign. Please double check and fix it."
>
> Thank you for your carefully reading and notification.
> You are right, and there is a missing minus sign in the proof of Theorem 3.1.
> We sincerely apologize for the oversight and we have **rectified the error**.
> To prevent such errors from cropping up again, we have conducted a thorough review of all the mathematical expressions and formulas throughout the paper, ensuring their accuracy and consistency.
>
> We truly appreciate your attention to detail. Thank you for helping us improve the precision and clarity of our work.
>
>
>
> > **Question 1:** - "The hardness is denoted as $\delta_j^{(u)}$ first and then $\delta_j^{(u,i)}$, which seems confusing. Is it related to a specific user u or a user-item pair (u, i)?"
>
> Thank you for highlighting this discrepancy in notation.
> You are right. The term $\delta_j^{(u,i)}$ denotes the hardness of a negative item $j$ with respect to observed interaction $(u,i)$.
> In the methodology section of our original paper, specifically around line 150, we simplified the notation to focus primarily on a single user-item pair $(u,i)$. This was done to facilitate ease of understanding for the reader. As we progressed further into the methodology, post line 177, we expanded our considerations to account for all observed interactions, thereby introducing the notation $\delta_j^{(u,i)}$.
>
> We acknowledge that this transitional shift in notation might cause confusion, and we truly appreciate your keen attention to detail. We will consider including an explanatory note or rephrasing to make this transition clearer in future versions.

---

> > ### Comment · Reviewer_PLbw · 2023-08-14
> >
> > Thank you for the classifications. My concerns have been resolved. BTW, I specifically appreciate the illustrative examples provided, which clearly present how the proposed model works. Given this, I increase my score slightly.

---

> > > ### Author Response · Authors · 2023-08-17
> > > **Thanks reviewer!**
> > >
> > > We sincerely thank you for recognizing our efforts in rebuttal. We appreciate your decision to increase the score. Your feedbacks have been invaluable to our work.

---

### Official Review · Reviewer_kSe8 · 2023-07-05

**Soundness:** 2 fair
**Presentation:** 2 fair
**Contribution:** 2 fair
**Rating:** 6
**Confidence:** 4

**Summary:**

Current losses for collaborative filtering struggle to handle the issue of unobserved user-item pairs.  Typically, the approach is to treat unseen pairs as negatives while seen pairs as positives, but this is somewhat problematic because unseen pairs could just be unobserved positives.  The authors propose an adversarial InfoNCE-based loss that claims to address this problem in collaborative filtering.  This loss works by minimizing the InfoNCE loss given that we have adversarially learned weights for the negative samples.  They give a theorem that shows this proposed loss can be interpreted as a distributionally robust optimization problem.  Finally, they give some empirical results showing the efficacy of their method over other CF baselines.

**Strengths:**

1.  This paper provides a method that appears to give solid gains across various collaborative filtering tasks.

2.  They do make an attempt to try to interpret what the \delta (adversarially learned parameters) in their method are doing.

3.  The graphs for the results and the pictures explaining the methods are good and helped me with understanding.

4.  The authors try to tackle a hard problem: it is hard to think about how to best utilize unseen pairs in collaborative filtering due to their unseen nature.

5.  The method is fairly novel as a nontrivial extension to InfoNCE to the collaborative filtering setting via adversarial training.

**Weaknesses:**

1.  I don't understand the role of the adversarial variables in the algorithm.  In figure 2a there was some attempt at interpreting the values of the deltas, but it still does not make sense.  I hope the authors can explain the role of the variables better.

2.  I think the terminology of "hard negative" is confusing, because typically in the self-supervised learning literature people call negatives that are near the decision boundary "hard negatives".  However, in this paper hard negatives are the opposite: negatives that are far away from the positives pair.  I suggest rewriting the paper to make the message more clear.

3.  In general, the paper is hard to understand and has many grammatical errors.  The authors should fix this to make the paper easier to read.

4.  From what I can understand, the loss should simply make the deltas as large as possible (positive) to increase the loss value given that there are no constraints (aside from the number of epochs trained, I guess).  I have concerns about the usefulness and stability of this algorithm.

5.  In theorem 3.1, we assume that the deltas imply a probability distribution.  Is this true?  As we train do the deltas for a user i add up to |N_i|?  I'm not sure there is a constraint there enforcing this.  In that case I'm not sure the theory applies to the algorithm as-is.

**Questions:**

My questions are in the weaknesses.

**Limitations:**

Seems to be sufficient.

---

> ### Author Rebuttal · Authors · 2023-08-10
>
> **Response to Reviewer $\color{red}{\text{kSe8}}$**
>
> We appreciate your comments, some of which inspires us to greatly improve our paper.
> Below we provide the point-to-point responses to address your concerns and clarify the misunderstandings of our proposed method.
> If you have additional questions, we would be pleased to discuss them with you.
>
> > **Comment 2: Terminology confusing** - "I think the terminology of "hard negative" is confusing, because typically in the self-supervised learning literature people call negatives that are near the decision boundary "hard negatives". However, in this paper hard negatives are the opposite: negatives that are far away from the positives pair. ..."
>
> We agree that hard negatives are negatives lying close to the decision boundary, making them hard to classify.
> However, a more capable model, exhibiting superior generalization, aims to reshape the representation space by pushing those hard negatives far from the decision boundary [1,2].
> Figure 1 in our original submission offers a visual explaination of this concept.
> It depicts the feasible zone for hard and false negatives of both InfoNCE and AdvInfoNCE.
> With the application of AdvInfoNCE, hard negatives are purposefully distanced from the positive pairs from a feasible zone perspective.
> By adjusting the feasible zone using $\delta$ in AdvInfoNCE, we effectively re-rank the negatives in representation space.
> We are confident in our choice of terminology and value further discussion if any points require elaboration.
>
>
> > **Comment 1: Role of $\delta$** - "I don't understand the role of the adversarial variables..."
>
> We value your comments. To better clarify the role of $\delta$ and its learning process for hard and false negatives, we **conducted additional illustrative examples**.
>
> The illustrative example should highlight two points. 1. AdvInfoNCE could effectively **identify the false and hard negatives** via learnable $\delta_j$ (max-stage). 2. $\delta$ helps to **refine the item ranking** compared to InfoNCE (min-stage).
>
> - **Identification of False and Hard Negatives:**
> On the Tencent training data, we trained both the InfoNCE and AdvInfoNCE models. Interactions unobserved during training but present in testing are labeled as false negatives (FN), otherwise true negatives (TN).
> Based on our theoretical assumption, for a FN, we wish a more relaxed constraint, leading to $\delta < 0$.
> To validate this assumption, we introduce the 'FN identification rate', a metric determining the proportion of FNs where $\delta < 0$.
> As Fig 6 in one-page uploaded pdf shows, our observations are consistent with our claim.
> As training proceeds, the FN identification rate increases, capping at nearly 70%.
> This reveals AdvInfoNCE's capability to identify approximately 70% of FNs in test set.
> We attribute the superior performance of AdvInfoNCE over InfoNCE to this gradual identification.
> - **Refinement of Item Ranking:**
> we randomly draw two users along with their FN and TN items, subsequently retrieving their associated $\delta$ values, ranking positions, and cosine similarities, as demonstrated in Fig 7.
> Consistent with our prior findings, for an FN, AdvInfoNCE generally assigns a negative $\delta$.
> This negative $\delta$, indicating a more lenient feasible zone constraint, enables the recommender to achieve higher cosine similarity.
> This, in turn, escalates the FN's ranking.
> For instance, as Fig 7(a) shows, given $\delta=−0.7887$, AdvInfoNCE elevates an FN from the 55th to a commendable 5th position. Conversely, for a TN, AdvInfoNCE leans towards a positive $\delta$, narrowing the feasible zone, thus distancing it from positives.
> An exemplary case is the TN $j_{6543}$ in Fig 7(a), where AdvInfoNCE, upon learning its $\delta = 1.1921$, declines its rank from 257th to 4587th.
> Such real-world cases attest to $\delta$'s role in fine-tuning recommendation ranking.
>
> In a word, for a specific u, the learnable $\delta_j$ measures the hardness of item j and further frame a fine-grained ranking criterion.
>
> > **Comment 3: hard to understand** -"In general, the paper is hard to understand ..."
>
> Thanks. We thoroughly proofread the manuscript and sincerely hope that our revised version will provide a smoother reading experience, aligning with the feedbacks we've received from other reviewers.
>
>
> > **Comment 4 & 5: Constraint of algorithm** - "From what I can understand, the loss should simply make the deltas as large as possible (positive) ..." "In theorem 3.1, we assume that the deltas imply a probability distribution. Is this true? ..."
>
> Thanks for your feedback.
> While we indeed delineate the constraints of AdvInfoNCE in various parts of our original manuscript, such as in lines 180, 187, 199-204, 547 and Eq 9, we recognize the need to highlight them more prominently.
>
> We define $\delta_j^{(u,i)}$ as $\log(|N_u|\cdot p(j|(u,i)))$, where $p(j|(u,i))$ denotes the probability of sampling negative item $j$ for observed interaction $(u,i)$.
> This signifies that the summation over all j for $p(j|(u,i))$ equals 1.
> Moreover, the bounds for $\delta_j^{(u,i)}$ are strictly set as $(\log(1-|N_u|\epsilon), \log(1+|N_u|\epsilon))$, which can be found in Eq 9.
>
> To provide an empirical insight into the $\delta$ value, we list its mean and standard deviation in our experiments during training: (-0.0, 0.0003), (-1e-4, 0.0164), (-0.0028, 0.0733), (-0.0128, 0.1549), (-0.0353, 0.2515), (-0.0747, 0.3572), (-0.1343, 0.4676).
> Noted that, the mean of $\delta$ is equivalent to $- D_{KL}(P_0||P)$, as stated in Line 195.
>
>
> [1] ArcFace: Additive Angular Margin Loss for Deep
> Face Recognition
>
> [2] Simplify and Robustify Negative Sampling for
> Implicit Collaborative Filtering

---

> ### Author Response · Authors · 2023-08-14
> **Follow-up discussion**
>
> Thank you for taking the time to review our paper. We appreciate your feedback and hope our response do address your concerns, especially regarding the terminology and the algorithm. We thus do hope our clarification of this main concern does help you reassess our paper. If you have additional concerns, we would be more than happy to provide additional clarification. Thank you for your attention.

---

> ### Comment · Reviewer_kSe8 · 2023-08-15
> **I raise my score.**
>
> Based on the rebuttals given by the authors and other reviews, I revise my score upwards.  I believe the findings in this paper would be a valuable contribution to the conference.
>
> I have one more question.  Why should we expect that a hard negative more likely correspond to a true negative at test time (as shown in Figure 7)?  I can see that the experiment results seem to support this, but why should this be intuitively true?

---

> > ### Author Response · Authors · 2023-08-16
> >
> > We would like to express our sincere appreciation for your review and for taking the time to reconsider our work based on the provided rebuttals.
> >
> > For your question regarding the hard negatives, our response is on the way. Thanks again.

---

> > ### Author Response · Authors · 2023-08-17
> > **Response to new questions**
> >
> > We appreciate you posing this insightful question regarding the relationship between true negatives and hard negatives.
> >
> > At its core, our hypothesis in AdvInfoNCE is grounded in this fundamental assumption about the relationship between true negatives and hard negatives.
> > Allow me to elucidate our rationale:
> >
> > 1. **Definition of Ture Negatives and Hard Negatives.**
> >
> > - **True Negatives:**
> > Within the recommendation framework of offline testing, negatives denote interactions absent from the training phase. While true negatives denote items that absent from both the training and testing phase and user intrinsically dislike, which is unknown.
> > The ultimate goal of offline recommendation is to discern potential interactions (positives) for testing based on the understanding gained from fitting the training data.
> > Equivalently speaking, offline recommendation testing is essentially to identify which items are true negatives: items that, even exposed to the user, are unlikely to be clicked on.
> > In other words, the recommender aims to avoid such true negatives appearing in the top rankings of the recommendation system.
> > - **Hard Negatives** learned by AdvInfoNCE: Hard negatives are interactions that are assigned a positive hardness value $\delta$ in AdvInfoNCE.
> > Such a value suggests a narrowing of the feasible zone, distancing the interaction's representations from positive.
> > In essence, AdvInfoNCE de-prioritizes these hard negatives in ranking.
> >
> > Hence, we expect the AdvInfoNCE can successfuly identify the true Negatives in dataset as its defined hard negatives.
> >
> > 2. **Hard negative mining justification.**
> > Here, we also want to elucidate why we view the process of identifying true negatives as akin to hard negative mining.
> > We define an item with $\delta > 0$ as a hard negative based on the gradient analysis presented in Appendix B.3.
> > The gradients associated with the negative item j are proportionally linked to $\exp(\delta_j)$.
> > In other words, for $\delta_j >0$, the recommender exhibits more attention to this item by a factor of $\exp(\delta_j) >1$.
> > This trait aligns with the concept of hard negatives in hard negative mining.
> >
> >
> > I hope this response offers a clearer, more concise understanding. Let me know if further clarifications are required.

---

> > > ### Comment · Reviewer_kSe8 · 2023-08-18
> > > **This still isn't clear to me.**
> > >
> > > I understand the AdvInfoNCE deprioritizes hard negatives and decreases their ranking.  But that is not necessarily the same as decreasing the ranking of true negatives is it?  Why should we expect that hard negatives are more likely to be true negatives?

---

> > > > ### Author Response · Authors · 2023-08-21
> > > > **Response to new questions**
> > > >
> > > > Thanks for your continued interest.
> > > > Your question seems to center on how our identified hard negatives are related to true negatives. If we have misunderstood, kindly clarify further.
> > > >
> > > > In our paper, we delve into the theoretical understanding of AdvInfoNCE's effectiveness using DRO, gradient analysis, and alignment with top-k evaluation metrics. Your question has prompted us to distill our method's findings in the most straightforward manner, and we have taken time to introspect to provide a reasonable response. We appreciate your patience and insightful question.
> > > >
> > > > Let us intuitively grasp our findings through the lens of item popularity.
> > > >
> > > > Many foundational papers acknowledge that sampled negative items of high popularity should be considered as true negatives [1,2].
> > > > Such papers indicate that when an item possesses high popularity but remains unclicked by a user, it implies a greater likelihood of the item being uninteresting to the user.
> > > > Concurrently, negatives with higher popularity are more likely to be identified as hard negatives by AdvInfoNCE.
> > > > During the "min" stage of AdvInfoNCE, popular items are prone to quickly achieving higher cosine similarity scores due to the intrinsic properties of CF backbones [3,4].
> > > > Consequently, during the "max" stage of adversarial training, popular items tend to incur increased penalties and higher hardness values compared to those unpopular items.
> > > > This phenomenon is also verified in the "averaged hardness" w.r.t. item popularity presented in Figure 2.(a) of our paper.
> > > >
> > > > In summary, items with higher popularity tend to be recognized as hard negatives by our method.
> > > > Furthermore, according to the preceding literature, these popular items might also qualify as potential true negatives.
> > > > This is how our identified hard negatives are related to true negatives.
> > > >
> > > > It is worth noting that, unlike false negatives, the true negatives here are unknown to us in fact.
> > > > False negatives pertain to interactions unobserved during training but present in testing. However, we lack insight into whether other negative samples, which do not appear in the test set, could indeed be true negatives.
> > > > Hence, we have come to realize that the two items in the case studies plotted on the right-hand side of Figure 7 can not be simply labeled as true negatives.
> > > > They should exclusively be referred to as sampled negatives since whether they are true negatives is unknown.
> > > > We have also revised the definition of true negatives as provided in the previous response.
> > > > From: "True negatives denote interactions absent from both the training and testing phases. " to "True negatives denote interactions absent from both the training and testing phases and which users inherently dislike, making them unknown from a data perspective."
> > > > Sorry for this confusion. We sincerely appreciate your valuable suggestions.
> > > > We hope that our answer resolves your question. If you have any further questions, we are happy to continue discussing with you.
> > > >
> > > > [1] Session-based recommendations with recurrent neural networks.
> > > >
> > > > [2] Fast Matrix Factorization for Online Recommendation with Implicit Feedback.
> > > >
> > > > [3] Debiasing Neighbor Aggregation for Graph Neural Network in Recommender Systems.
> > > >
> > > > [4] Popularity-Opportunity Bias in Collaborative Filtering.

---

> > > > > ### Comment · Reviewer_kSe8 · 2023-08-21
> > > > >
> > > > > Yes, thank you for making this clear.  Your assumption that popular non-clicked items are more likely to be true negatives is essentially the core of the paper and I suggest it be emphasized in future drafts.

---

> > > > > > ### Author Response · Authors · 2023-08-22
> > > > > > **Thanks!**
> > > > > >
> > > > > > Thank you for your insights and questions. Your feedbacks have been invaluable to our work.

---

### Official Review · Reviewer_ikze · 2023-07-08

**Soundness:** 3 good
**Presentation:** 3 good
**Contribution:** 3 good
**Rating:** 7
**Confidence:** 4

**Summary:**

This paper proposes a principled AdvInfoNCE loss for CF methods to improve generalization ability. It utilizes a fine-grained hardness-aware ranking criterion to assign weights for unobserved user-item interactions. In this way, it can enable better distinction between different negative, thus mitigating the inductive bias in CF-based methods. It provides theoretical proof of the effectiveness of AdvInfoNCE loss and the experimental results compared with other popular loss used in recommenders look promising.

**Strengths:**

It is well-written and easy to follow.
Good motivation of improving the generalizability of CF-based methods.
It provides thereotical guarantees for the loss design and conducts comprehensive analysis on the effectiveness of its method.
Experimental results compared with other popular functions adopted in CF models look promising.
Code is open.


**Weaknesses:**

Experiments can be more extensive. The results on MF and LightGCN look promising. But I think it would be more convincing if the authors can consider more CF-based backbones like MultVAE [1] and DGCF [2].

[1] Liang et al. Variational Autoencoders for Collaborative Filtering. 2018 WWW.
[2] Wang et al. Disentangled Graph Collaborative Filtering. 2020 SIGIR.

**Questions:**

This paper proposes a novel constrastive loss for fine-grained ranking of negative samples to improve the generalizability of CF-based models, which is a significant contribution. It provides comprehensive theoretical analysis and experiments look promising. I just have minor concerns about the technical details.
- Do you attempt other similarty measurements for calculating the hardness and whether the choice of similarity calculation matters?
- What is the rate of negative samples over all the unobserved interactions used in your method and is there any  result of using different negative sampling rate?

**Limitations:**

More CF-based backbones can be considered.

---

> ### Author Rebuttal · Authors · 2023-08-10
>
> **Response to Reviewer $\color{blue}{\text{ikze}}$**
>
> We sincerely thank you for your time and valuable comments. Your main suggestions about considering additional CF-backbones help us substantiate wide applicability of AdvInfoNCE.
>
> > **Comment 1: Additional CF-based backbones** - "Experiments can be more extensive. ..."
>
> Thanks for your great suggestions!
> We fully agree that considering a broader range of CF-based backbones will better showcase the applicability of AdvInfoNCE loss.
> While the rebuttal period is time-constrained, we have incorporated the AdvInfoNCE loss to both a GCN-based CF backbone: UltraGCN [3], and a VAE-based CF backbone: VGAE [4].
>
> We deeply appreciate your suggestion of considering DGCF [2] and MultVAE [1].
> As outlined in the DGCF paper, it fundamentally emerges as a special case of LightGCN under the multi-intent assumption. Given the extensive experiments we have already conducted with LightGCN, we chose UltraGCN which offers an ultra-simplified
> formulation beyond LightGCN.
> As for MultVAE, it primarily takes into account the user-by-item interaction matrix instead of traditional user/item embeddings.
> Adapting AdvInfoNCE to MultVAE requires re-defing interaction-wise negative sampling and restructuring the entire training pipeline, which we leave for future work.
> What we accomplished during the rebuttal period was to demonstrate how AdvInfoNCE can be applied to another VAE-based backbone, VGAE.
>
> To demonstrate the applicability of AdvInfoNCE under UltraGCN and VGAE, we show the results on Tencent in Table 1. Clearly, AdvInfoNCE boosts the recommendation performance of UltraGCN and VGAE in terms of various OOD settings by a large margin. More detailed analyses can be found in our revision paper in Appendix.
>
> **Table 1: Overall performance for UltraGCN and VGAE backbones**
> |                 |          | $\gamma=200$ |       |       | $\gamma=10$ |      |       | $\gamma=2$ |        | Validation|
> | :-------------: | :------: | :-----: | :-----: | :-----: | :-----: | :-----: | :-----: | :-----: | :-----: |:-----: |
> | | HR | Recall | NDCG | HR | Recall | NDCG | HR | Recall | NDCG |NDCG|
> | UltraGCN | 0.0930 | 0.0343 | 0.0190 | 0.0567 | 0.0215 | 0.0119 | 0.0400 | 0.0157 | 0.0095 | 0.0682 |
> |UltraGCN+ InfoNCE | 0.1436 | 0.0519 | 0.0303 | 0.0896 | 0.0324 | 0.0189 | 0.0617 | 0.0227 | 0.0135 | 0.0842 |
> |UltraGCN+ AdvInfoNCE | $\underline{0.1538}$ | $\underline{0.0569}$ |$\underline{0.0338}$ | $\underline{0.1025}$ | $\underline{0.0380}$ | $\underline{0.0227}$ |$\underline{0.0726}$ | $\underline{0.0276}$ | $\underline{0.0168}$ | 0.0883 |
> |  VGAE + InfoNCE   |   0.1482   |    0.0536    |   0.0315   |   0.0923   |   0.0338    |   0.0202   |   0.0640   |   0.0237   |   0.0141   |   0.0823   |
> | VGAE + AdvInfoNCE | **0.1588** |  **0.0589**  | **0.0353** | **0.1069** | **0.0395**  | **0.0239** | **0.0778** | **0.0296** | **0.0182** | 0.0871 |
>
> > **Question 1: Different similarity measurements** - "Do you attempt other similarty ...?"
>
> Thank you for highlighting this question. In light of this, we **conducted new experiments** evaluating both inner product and cosine similarity measurements for hardness calculation. As indicated in Table 2, the choice between these measurements doesn't introduce significant discrepancies in performance.
>
> **Table 2: Varying similarity measurements**
> |                          |            | $\gamma=200$ |            |            | $\gamma=10$ |            |            | $\gamma=2$ |            | Validation |
> | :----------------------: | :--------: | :----------: | :--------: | :--------: | :---------: | :--------: | :--------: | :--------: | :--------: | :--------: |
> |                          |     HR     |    Recall    |    NDCG    |     HR     |   Recall    |    NDCG    |     HR     |   Recall   |    NDCG    |    NDCG    |
> |    AdvInfoNCE-cosine     |   0.1561   |    0.0581    |   0.0346   |   0.1046   |   0.0389    |   0.0237   |   0.0750   |   0.0286   |   0.0175   | **0.0881** |
> | AdvInfoNCE-inner product | **0.1600** |  **0.0594**  | **0.0356** | **0.1087** | **0.0403**  | **0.0243** | **0.0774** | **0.0295** | **0.0180** |   0.0879   |
>
>
> > **Question 2: Different negative sampling rate** - "What is the rate ...?"
>
> Your point on negative sampling rate is noteworthy.
> The detailed information of negative samples on different datasets can be found in Appendix (Table 9).
> As expounded in Theorem 3.1, our AdvInfoNCE design capitalizes on obtaining high-quality hard negative samples from a distribution perspective. Theoretically, increasing the number of negative samples should yield a more favorable negative distribution. However, the trade-off with computational expense led us to adopt 128 negative samples for all Tencent baseline methodologies. In response to your query, we evaluated two other rates (64 and 256 samples) as represented in Table 3. Our findings consistents with our initial thinking: higher negative sampling rates do offer performance enhancements for AdvInfoNCE.
>
>
> **Table 3: Varying Number of Negative Sampling on Tencent**
> |      |      | $\gamma=200$ |      |      | $\gamma=10$ |      |      | $\gamma=2$ |      | Validation |
> | :--: | :--: | :----------: | :--: | :--: | :---------: | :--: | :--: | :--------: | :--: | :--------: |
> |      |  HR  |    Recall    | NDCG |  HR  |   Recall    | NDCG |  HR  |   Recall   | NDCG |    NDCG    |
> |  64  | 0.1513 | 0.0563 | 0.0333 | 0.1006 | 0.0373 | 0.0225 | 0.0708 | 0.0269 | 0.0164 | 0.0854 |
> | 128  | 0.1600 |  0.0594  | 0.0356 | 0.1087 | 0.0403  | 0.0243 | 0.0774 | 0.0295 | 0.0180 |   0.0879   |
> | 256  | **0.1642** | **0.0609** | **0.0367** | **0.1125** | **0.0419** | **0.0253** | **0.0815** | **0.0310** | **0.0189** | **0.0889** |
>
> [1] Variational Autoencoders for Collaborative Filtering. 2018
>
> [2] Disentangled Graph Collaborative Filtering. 2020
>
> [3] UltraGCN: Ultra Simplification of Graph Convolutional Networks for Recommendation. 2021
>
> [4] Variational Graph Auto-Encoders. 2016

---

> > ### Comment · Reviewer_ikze · 2023-08-21
> >
> > Thank you for the efforts. My concerns have been addressed and I will maintain my rating.

---

### Author Rebuttal · Authors · 2023-08-10

We are delighted to see the contributions of our paper have been acknowledged by the majority of the Reviewers. Specifically, we appreciate the Reviewers' recognition of our motivation, theoretical analysis ($\color{blue}{\text{ikze}}$, $\color{orange}{\text{PLbw}}$, $\color{purple}{\text{qmEP}}$, $\color{green}{\text{1hVj}}$), and novelty ($\color{blue}{\text{ikze}}$ , $\color{red}{\text{kSe8}}$, $\color{green}{\text{1hVj}}$).


We appreciate all the reviewers for their valuable comments and suggestions. This helped improve our submission and better strength our claims. Taking into account suggestions of Reviewers, we have summarized the updates to the paper as follows:

- **More detailed explanations.** Addressing the concerns raised by Reviewers $\color{red}{\text{kSe8}}$ and $\color{green}{\text{1hVj}}$, we have incorporated **two additional illustrative experiments** on the dynamic evolution of the fine-grained hardness $\delta_{j}$ and case studies explaining the effect of $\delta_{j}$.

- **Experiments on two collaborative filtering backbones.** Following the suggestions of Reviewer $\color{blue}{\text{ikze}}$, we have conducted experiments on **two new CF backbones** to validate the generalization ability of AdvInfoNCE.

- **Comparison experiments.** In response to Reviewers $\color{purple}{\text{qmEP}}$ and $\color{green}{\text{1hVj}}$, we have added literature reviews and **four supplementary comparison experiments** in the fields of negative sampling, adaptive SSL, and recommendation debiasing.

- **Details about AdvInfoNCE.** We have incorporated detailed discussion about AdvInfoNCE, including hyperparameters selection, instability and computational complexity, to address the concerns of Reviewers $\color{blue}{\text{ikze}}$, $\color{orange}{\text{PLbw}}$ and $\color{purple}{\text{qmEP}}$.

We have tried our best to address the main concerns raised by reviewers and we hope that these improvements will be taken into consideration.
We also present the point-to-point responses for each reviewer below.

---

### Decision · Program_Chairs · 2023-09-21

**Decision:**

Accept (poster)

**Comment:**

The paper introduces a novel adversarial contrastive loss, Adversarial InfoNCE, designed to enhance the performance, particularly in out-of-distribution (OOD) scenarios, of collaborative filtering in top-k recommendation systems. Despite initial concerns raised during the review process, the majority were resolved satisfactorily. All reviewers collectively, with some expressing strong support, recommended the acceptance of the paper. It is highly advised that the subsequent revision includes the additional discussions and results discussed during the review process.